# Overestimated nitrogen loss from denitrification for natural terrestrial ecosystems in CMIP6 Earth System Models

Maoyuan Feng [1,2], Shushi Peng [1,2] ✉, Yilong Wang[3], Philippe Ciais [4,5], Daniel S. Goll[4], Jinfeng Chang [6], Yunting Fang [7], Benjamin Z. Houlton[8], Gang Liu[1,2], Yan Sun[9] & Yi Xi [1,4]

Denitrification and leaching nitrogen (N) losses are poorly constrained in Earth System Models (ESMs). Here, we produce a global map of natural soil $^{15}$N abundance and quantify soil denitrification N loss for global natural ecosystems using an isotope-benchmarking method. We show an overestimation of denitrification by almost two times in the 13 ESMs of the Sixth Phase Coupled Model Intercomparison Project (CMIP6, $73 \pm 31$ Tg N yr$^{-1}$), compared with our estimate of $38 \pm 11$ Tg N yr$^{-1}$, which is rooted in isotope mass balance. Moreover, we find a negative correlation between the sensitivity of plant production to rising carbon dioxide ($CO_2$) concentration and denitrification in boreal regions, revealing that overestimated denitrification in ESMs would translate to an exaggeration of N limitation on the responses of plant growth to elevated $CO_2$. Our study highlights the need of improving the representation of the denitrification in ESMs and better assessing the effects of terrestrial ecosystems on $CO_2$ mitigation.

Nitrogen (N) is a crucial nutrient that regulates plant growth and its response to elevated carbon dioxide ($CO_2$) concentration in a wide range of terrestrial ecosystems[1–3]. Incorporating the interactions between nitrogen (N) and carbon (C) into Earth System Models (ESMs) helps improve future projections of the coupled carbon-climate system[4–6]. In the fifth Phase of the Coupled Model Intercomparison Project (CMIP5) for the IPCC AR5 report, the terrestrial N cycle was represented in only two ESMs (CESM and NorESM)[7], both of which relied on the same land surface model (Community Land Model, CLM) to estimate N cycle interactions. This land surface model was found to include an unrealistic representation (coarse overestimation) of the N losses from the denitrification pathway[8–10]. In the most recent

assessment (CMIP6), 24 out of 44 ESMs include the N cycle, but rely on different assumptions/theories for relevant processes[6,7,11,12]. It remains unclear whether current ESMs have improved the N loss estimates compared to the CMIP5 models.

Nitrogen losses are pivotal in determining N availability for plants and microbes[3,13,14]. However, the evaluation of N loss fluxes is challenging due to the difficulties of measuring the denitrified dinitrogen ($N_2$) emissions directly[15,16] and scaling up point scale observations to global fields[9,17,18]. The natural N isotope ratio ($^{15}$N/$^{14}$N or $\delta^{15}$N) in soil is an important indicator for partitioning gaseous N losses (denitrification and volatilization) from aquatic (leaching) ones, since the former pathway has a much stronger discrimination against the heavier

[1]Sino-French Institute for Earth System Science, College of Urban and Environmental Sciences, and Laboratory for Earth Surface Processes, Peking University, Beijing, China. [2]Institute of Carbon Neutrality, Peking University, Beijing, China. [3]State Key Laboratory of Tibetan Plateau Earth System, Resources and Environment, Institute of Tibetan Plateau Research, Chinese Academy of Sciences, Beijing, China. [4]Laboratoire des Sciences du Climat et de l'Environnement, LSCE/IPSL, CEA-CNRS-UVSQ, Université Paris-Saclay, Gif-sur-Yvette, France. [5]The Cyprus Institute 20 Konstantinou Kavafi Street, 2121 Nicosia, Cyprus. [6]College of Environmental and Resource Sciences, Zhejiang University, Hangzhou, China. [7]CAS Key Laboratory of Forest Ecology and Management, Institute of Applied Ecology, Chinese Academy of Sciences, Shenyang, China. [8]Department of Ecology and Evolutionary Biology and Department of Global Development, CALS, Cornell University, Ithaca, NY, USA. [9]College of Marine Life Sciences, Ocean University of China, Qingdao, China. ✉ e-mail: speng@pku.edu.cn

isotope ($^{15}$N)[19,20]. Using a simple global map of soil $\delta^{15}$N upscaled from ~50 observations and linear relationships with climate drivers[21], Houlton et al[8]. utilized a framework of isotope mass-balance equations to constrain the ratio ($f_{denit}$) of N loss from denitrification relative to total N losses, and found that the Community Land Model (CLM-CN) coarsely overestimated both the pattern and magnitude of $f_{denit}$. Thousands of soil $\delta^{15}$N observations have occurred since the first global soil $\delta^{15}$N map from Amundson et al.[21] was published in 2003. These new observations can be leveraged to improve the quality of both the global soil $\delta^{15}$N map and $f_{denit}$. Moreover, applying a machine learning method and accounting for additional predictors, including climate drivers, microbial associations[22,23] and soil properties[24,25], has been shown to improve continental-scale soil $\delta^{15}$N estimates (e.g., in South America, by Sena-Souza et al.[26]), and is therefore expected to further improve the reliability of the global soil $\delta^{15}$N map[27].

In this study, our objective is to improve the current isotope benchmarking technique by deriving a spatial distribution of $f_{denit}$ estimates from soil $\delta^{15}$N observational data coupled with machine learning, and then use the model to constrain denitrification N losses as simulated by the CMIP6 models. First, we use 5887 direct measurements of soil $\delta^{15}$N in natural ecosystems from the literature[26,28] (see Methods; Supplementary Fig. 1), and produce a global soil $\delta^{15}$N map at a spatial resolution of 0.1° × 0.1° by using a Random Forest (RF) model (see Methods and Supplementary Text 1; Supplementary Fig. 2). This global soil $\delta^{15}$N map is used to benchmark the global map of $f_{denit}$ using isotope mass balance equations proposed by Houlton et al.[19] and Houlton and Bai[20] (Supplementary Texts 2–4). With the global map of isotope-benchmarking based $f_{denit}$, we then estimate the denitrification N loss of global natural terrestrial ecosystems under steady state (total N losses equal to total N inputs) and non-steady state with the

total N losses simulated by the CMIP6 ESMs. Our results indicate that the CMIP6 models substantially overestimate denitrification N losses.

## Results and discussion

### A global map of isotope-benchmarking based $f_{denit}$

We derived a global soil $\delta^{15}$N map using a robust RF model (Fig. 1a, see Methods and Supplementary Text 1), which performs well in capturing the nonlinear relationships between soil $\delta^{15}$N observations and predictors ($R^2 = 0.92$, Root Mean Square Error (RMSE) = 0.77‰) and also in predicting the soil $\delta^{15}$N ($R^2 = 0.55$, RMSE = 1.83‰) (Supplementary Text 1; Supplementary Figs. 2 and 3). The soil $\delta^{15}$N map has a global mean of 4.8‰ weighted by grid level N input (proportional to soil N content at steady state, and estimated as the product of N input flux and grid area) (Fig. 1a; Supplementary Fig. 4), which is slightly lower than the previous estimate of 5.5‰[8,21]. The spatial pattern of the soil $\delta^{15}$N map indicates a decreasing trend from low to high latitudinal regions, resulting in a latitudinal gradient of −0.5‰ per 10° increase in latitude. Compared to the soil $\delta^{15}$N map produced by a linear regression model by Amundson et al.[21], our RF model greatly increased the $R^2$ between observations and predictions across 933 grid cells from 0.20 to 0.93 and decreased the RMSE from 2.82‰ to 0.77‰ (Supplementary Fig. 5).

Based on the isotope balance equations of Houlton et al.[19] and Houlton and Bai[20] (see Methods), we used the global soil $\delta^{15}$N map to benchmark the $f_{denit}$ (Fig. 1b, Supplementary Text 2). The isotope-based $f_{denit}$ relies on the relative fractions of N inputs from rock N weathering, N deposition, and biological nitrogen fixation (BNF) that have contrasting $\delta^{15}$N signals[15]. To account for the uncertainties in these N input data, we derived an ensemble of global maps of isotope-based $f_{denit}$ using six sets of N inputs by combining a global map of rock

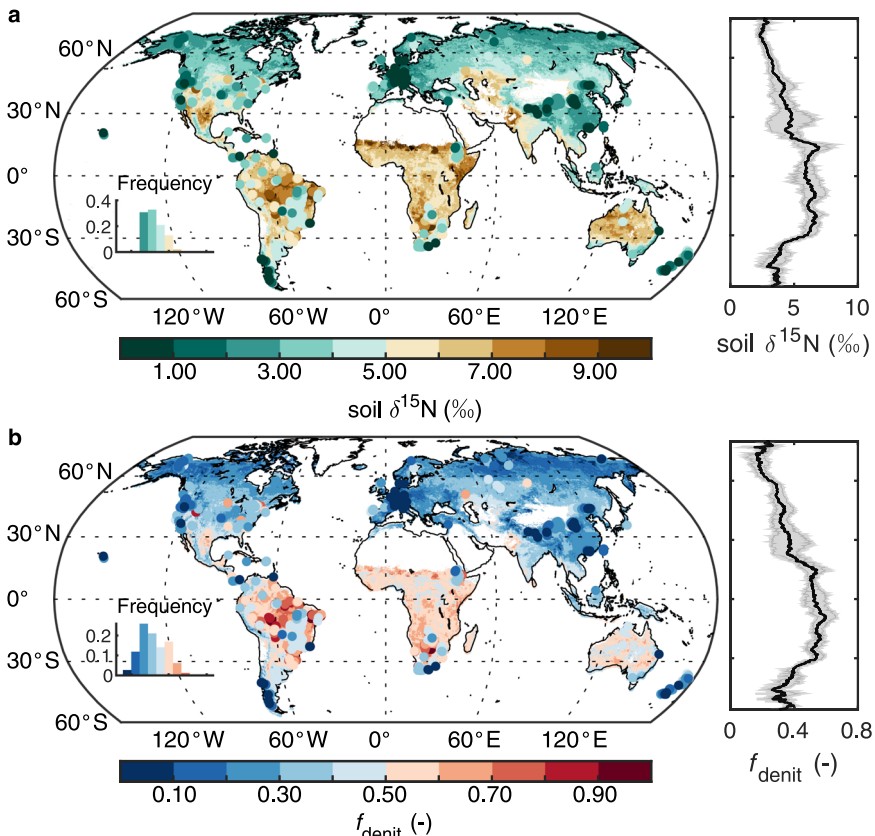

**Fig. 1 | Global maps of the soil $\delta^{15}$N and the isotope-benchmarking based fraction of denitrification N loss ($f_{denit}$) in natural terrestrial ecosystems.** **a** Global map of the mean of soil $\delta^{15}$N produced by Random Forest models. **b** Global map of the ensemble mean of $f_{denit}$ derived using six sets of N inputs. The colored dots represent the field measured soil $\delta^{15}$N and corresponding $f_{denit}$ in (**a**) and (**b**), respectively. Note that these two maps are upscaled from climate, soil and microbial symbionts, and other predictors (Supplementary Table 7) only for natural ecosystems.

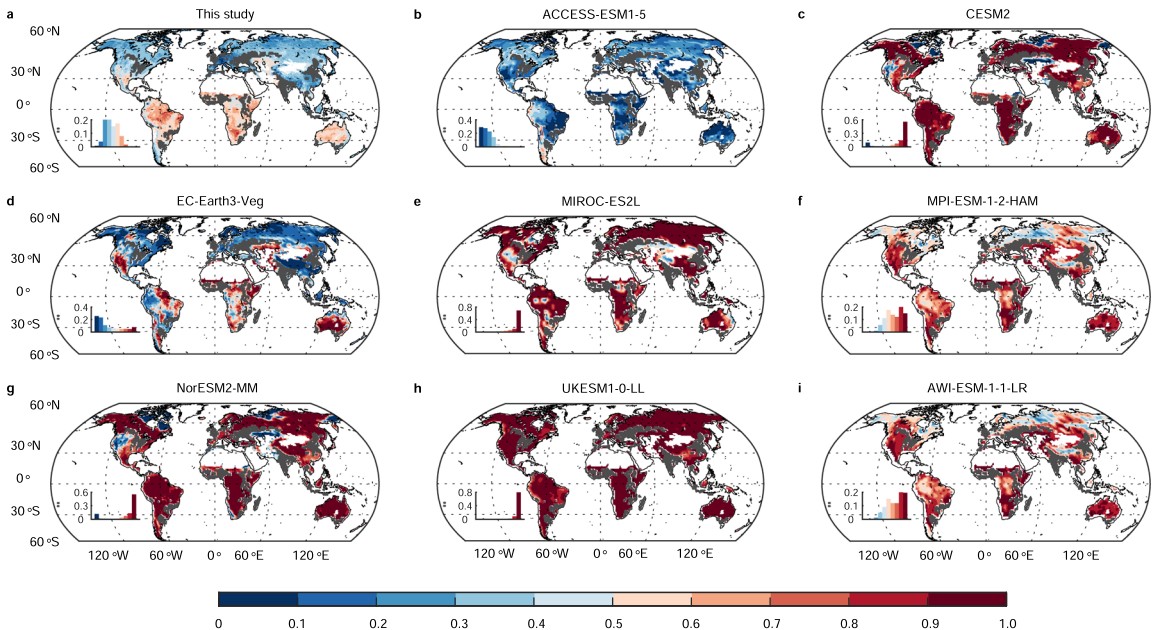

**Fig. 2 | Comparison of global maps of the fraction of denitrification N loss ($f_{denit}$) simulated by CMIP6 Earth System Models (ESMs) with the isotope-benchmarking based estimate of this study. a** Global map of the isotope-benchmarking based $f_{denit}$ of this study. **b–i** Global maps of $f_{denit}$ during the period 2005–2014 simulated by CMIP6 ESMs, with each representing a family of ESMs with similar patterns of $f_{denit}$. In each panel, the histogram in the bottom left corner shows the frequency distribution of $f_{denit}$ values across the globe. All crop and pastural areas were excluded from the analysis and are represented by grey regions.

N weathering[29] (10 Tg N yr$^{-1}$ with a global mean $\delta^{15}$N of 4.02‰), two global maps of N deposition[30,31] (an average of 40 Tg Nyr$^{-1}$ with a constant $\delta^{15}$N of 0‰) and three global maps of BNF[32] (an average of 57 Tg Nyr$^{-1}$ with a constant $\delta^{15}$N of −2‰) (see Methods and Supplementary Text 2). The ensemble of global maps of isotope-based $f_{denit}$ are similar, providing a global N input weighted average of $f_{denit}$ equal to 0.42 ± 0.01 (mean ± standard deviation (SD)) (Supplementary Table 1), i.e., 42 ± 1% of N losses in natural ecosystems occur via the denitrification pathway, slightly higher than previous area-weighted estimates of 26–40% under similar isotope-based framework but with a global map of soil $\delta^{15}$N from Amundson et al.[21] and different parameterizations[8,19,20]. Here, we show the isotope-benchmarking based $f_{denit}$ as the ensemble mean derived from the six sets of N inputs (107 ± 19 Tg N yr$^{-1}$, with a $\delta^{15}$N of −0.66 ± 0.21‰) (Fig. 1b), with detailed ensembles of global $f_{denit}$ maps presented in the Supplementary Information (Supplementary Text 2; Supplementary Fig. 6). Spatially, the isotope-benchmarking based $f_{denit}$ decreases from low to high latitudinal regions, with a latitudinal gradient of −0.05 per 10° latitude increase (Fig. 1a). In Amazonia and South Africa, the $f_{denit}$ is higher than 0.6, while in most grid cells over mid- and high latitude regions $f_{denit}$ is lower than 0.3. This spatial pattern is consistent with previous isotope-based studies (Supplementary Fig. 7)[6,8], and empirical knowledge, indicating a more open nitrogen cycle in the tropics compared with the boreal regions[1,2,13]. The uncertainty (quantified by standard deviations, SDs) of the isotope-based $f_{denit}$ across the six maps (Supplementary Fig. 8b) is much lower than the uncertainty from the benchmarking $f_{denit}$ that is propagated from the map of soil $\delta^{15}$N (Supplementary Fig. 8a).

## Large discrepancy between isotope-benchmarking based $f_{denit}$ and ESMs

Our findings reveal a large discrepancy between the isotope-benchmarking based $f_{denit}$ and the values simulated by the CMIP6 ESMs (Fig. 2 and Supplementary Fig. 9). In most of these ESMs (except ACCESS-ESM1-5 and EC-Earth3-Veg), $f_{denit}$ is relatively uniform across the globe and follows a highly skewed, roughly binary distribution, i.e., >90% of grid cells are at ~1 and the remaining <10% of grid cells are at

~0, resulting in an overestimated $f_{denit}$. The overestimation of $f_{denit}$ is in line with the previous isotope-based analysis[8] and observation based comparisons[9,10]. Moreover, the overestimated $f_{denit}$ is also found in CESM[8–10], one out of the two CMIP5 ESMs that included nitrogen-carbon interactions, suggesting little improvement has been made in the representation of denitrification in this ESM. Note that the isotope-benchmarking based $f_{denit}$ is sensitive to the isotope effect of denitrification ($\varepsilon_{denit}$), which has been reported to have large variations, i.e., 10–20‰ in natural soil communities[20] and 31–65‰ in pure incubation in the laboratory[33,34]. As $\delta^{15}$N observations in natural soil were collected in this study, following Houlton and Bai[20] and Houlton et al.[8], we adopted a value for the isotope effect of denitrification ($\varepsilon_{denit}$) of 13‰, at the lower end of previously reported values[20], resulting in a conservative (high) estimate of $f_{denit}$ (see Methods). If a higher $\varepsilon_{denit}$ had been adopted, the isotope-based $f_{denit}$ would have been even lower, pointing to an even more substantial overestimation of $f_{denit}$ in the CMIP6 ESMs (Supplementary Text 3; Supplementary Table 2; Supplementary Fig. 10).

We found that ESMs with a higher global mean estimate of $f_{denit}$ had a higher fraction of grid cells with $f_{denit} \approx 1$, the upper bound (Supplementary Fig. 11). Thus, most of the ESMs with overestimates of $f_{denit}$ are likely to be constrained by the upper bound and show small seasonal and interannual variations of $f_{denit}$ (Supplementary Figs. 12–14), which is contradictory to the empirical knowledge that the fraction of N loss from denitrification is highly dependent on the temperature and soil moisture[35–37]. Moreover, highly overestimated values of $f_{denit}$ ($\approx 1$) also imply that the N leaching losses were close to zero, which runs counter to the observation that dissolved N losses contribute substantially to N balances in many terrestrial ecosystems[16,38,39]. These contradictions suggest that denitrification, or related N cycle processes, are still poorly represented in the CMIP6 ESMs (Supplementary Text 1). Theoretically, denitrification rates depend on nitrogen and carbon availability, temperature, soil moisture, pH, and other factors[35,40,41]. We summarized the representations of denitrification and leaching N losses in the CMIP6 ESMs (Supplementary Table 3; Supplementary Text 5), and found that five families of models (CESM2, NorESM2, AWI-ESM1, MPI-ESM, and MIROC)

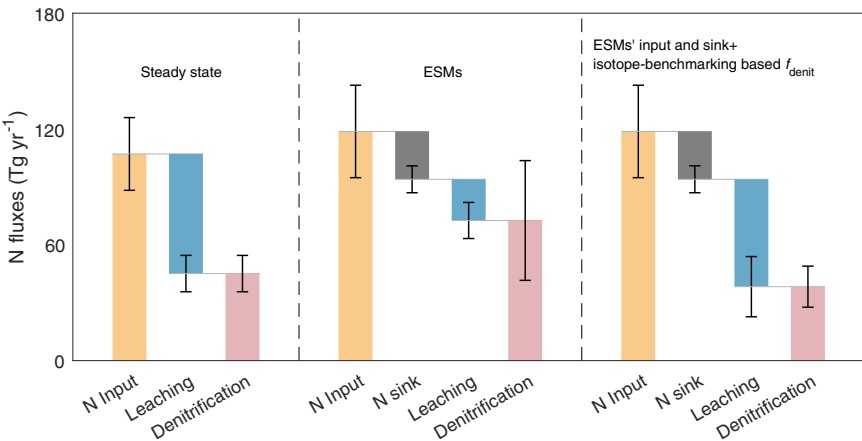

**Fig. 3 | Synthesis of nitrogen fluxes over global natural terrestrial ecosystems.** The denitrification N loss at steady state was estimated as the product of the isotope-benchmarking based $f_{denit}$ and total N losses that were assumed to be equal to the six sets of N inputs (atmospheric N deposition, biological nitrogen fixation (BNF), and rock N weathering; Supplementary Table 1). The error bars on the steady-state N fluxes are standard deviations derived from the six sets of N inputs

(Supplementary Table 1). The N fluxes simulated by the Earth System Models (ESMs) were directly downloaded from CMIP6 (https://esgf-node.llnl.gov/search/cmip6/). The third group of N fluxes retain the ESMs' N inputs and sinks, but use the isotope-benchmarking based $f_{denit}$ to re-allocate the total N losses simulated by the ESMs. The error bars on the N fluxes are standard deviations across the 13 ESMs (Supplementary Table 6). Source data are provided in Source Data file.

simulated the denitrification rate as a product of the simulated soil mineral N pool by using a scaling factor function of environmental variables. Three families of models (ACCESS-ESM1-5, EC-Earth3, and UKESM1) assume that the denitrification is a fraction of net or gross mineralization rates. Both groups of ESMs overestimated $f_{denit}$ except for EC-Earth3-Veg in which the denitrification rate is simulated as 1% of the gross mineralization rate[5]. Thus, in EC-Earth3-Veg, the spatial pattern, a decreasing latitudinal gradient of $f_{denit}$ from tropical to boreal regions, essentially reflects that of gross mineralization rate. Overall, the representation of denitrification processes should be improved by accounting for recent advances in theoretical understanding and data availability related to currently omitted but crucial processes in the nitrogen cycle, such as N-related microbial processes[42,43], retention of reduced and oxidized N form[44], and interactions between plant and soil microbes[45].

## Overestimated denitrification N loss from global natural ecosystems in CMIP6 ESMs

By applying the isotope-benchmarking based $f_{denit}$ under the steady-state assumption (total N losses equal to total N inputs), we estimated the denitrification N loss from global natural terrestrial ecosystems as $45 \pm 9$ Tg yr$^{-1}$ (Fig. 3; Supplementary Table 1; Supplementary Table 4; Supplementary Fig. 15) using six sets of global N inputs ($107 \pm 19$ Tg Nyr$^{-1}$), close to the recent estimates of 44–47 Tg Nyr$^{-1}$ using similar isotope based framework but with different parameters and global N inputs[15,20]. In recent decades, natural terrestrial ecosystems have acted as a N sink due to the accumulating terrestrial carbon sink[46,47]. Thus, the actual denitrification N loss in recent decades should be much lower than our steady-state estimate ($45 \pm 9$ Tg Nyr$^{-1}$). Across the 13 CMIP6 ESMs, the mean denitrification N loss is $73 \pm 31$ Tg Nyr$^{-1}$, with the mean of the terrestrial N sinks (vegetation plus soil N sinks) being $25 \pm 7$ Tg N yr$^{-1}$ and the mean of the N inputs being $119 \pm 24$ Tg Nyr$^{-1}$ (Fig. 3; Supplementary Table 5). With these N inputs and sinks from the ESMs, we utilized the isotope-benchmarking based global map of $f_{denit}$ to estimate the denitrification N loss as $38 \pm 11$ Tg Nyr$^{-1}$ (Fig. 3; Supplementary Table 6), considering that the effect of the terrestrial N sink on the isotope-based $f_{denit}$ is very limited (< 1%; see Methods and Supplementary Text 6; Supplementary Fig. 16). This calculation suggests that the CMIP6 ESMs overestimate the denitrification N loss by 92%, which would further bias the atmospheric chemistry (e.g., atmosphere N$_2$O, NO and NO$_2$ concentrations and

attendant chemical processes) if resolved in the ESMs. Conversely, the CMIP6 ESMs underestimate the leaching N loss by 62% (Fig. 3), which implies underestimated N loads to global aquatic ecosystems and the ocean, and consequently underestimate eutrophication in aquatic ecosystems and ocean productivity in the models. Note that the model bias is defined as the difference between isotopically constrained estimates and ESMs' simulated values, which provides an approximation of the true model bias as we lack direct observations of N losses at global scale. Our results suggest that the denitrification and leaching N losses in ESMs should be cross-constrained by $\delta^{15}$N data and N flux in stream and river discharges before using ESMs to study the N cycle between land and the ocean/atmosphere. Moreover, the responses of total N losses (denitrification plus leaching) to future climate change will be biased in the CMIP6 ESMs: a bias which could further propagate into the CMIP6 simulations of carbon-climate feedback[7].

## Exaggerated N limitation on plant growth due to overestimated denitrification N losses

Under elevated levels of atmospheric CO$_2$, nitrogen losses affect the occurrence of N limitation for the plant growth in natural ecosystems by controlling the rate at which the soil N availability changes over time[6]. Thus, we hypothesized that the overestimated denitrification N losses in ESMs lead to an underestimation of soil N availability and a further exaggeration of N limitations on the responses of plant growth to elevated CO$_2$ levels. We used a parameter $\beta_{NPP}$ to quantify the sensitivity of Net Primary Production (NPP) to elevated atmospheric CO$_2$ concentration using a regression approach from the historical simulations of ESMs (see Methods). We found a negative correlation between $\beta_{NPP}$ and $f_{denit}$ across 10 ESMs in boreal regions (50°–90°N) where N availability was generally low during the period 1960–2014 (Fig. 4, $R^2 = 0.69$, $p = 0.003$). In other words, an ESM with a higher $f_{denit}$ is more likely to underestimate $\beta_{NPP}$ and exaggerate the effect of N limitation on plant productivity. In boreal regions, compared to the isotope-benchmarking based $f_{denit}$ ($0.23 \pm 0.05$), the value of $f_{denit}$ is on average overestimated by 170% in the ESMs ($0.62 \pm 0.28$). Considering an increase of CO$_2$ concentration of 81 ppm during the period 1960–2014, the overestimation of $f_{denit}$ results in an underestimation of $\beta_{NPP}$ by the ESMs of 0.07% ppm$^{-1}$ which corresponds to 6% of the NPP increase in boreal regions. Our results highlight that ESMs exaggerate the N limitation on the responses of plant growth to elevated CO$_2$ in boreal regions. The exaggeration of the N limitation in ESMs

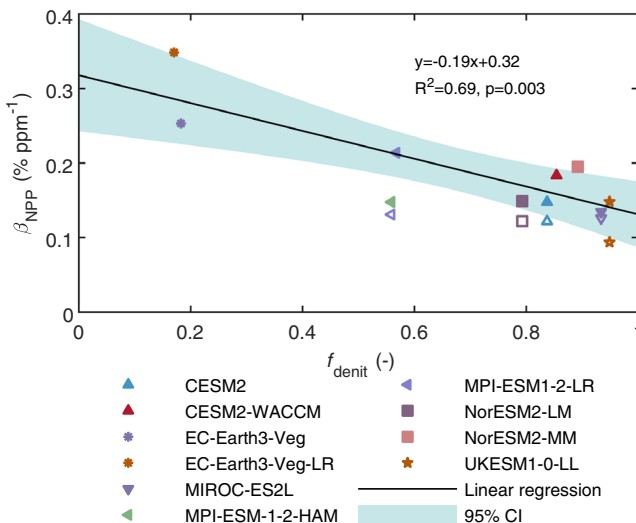

**Fig. 4 | Negative correlation between the parameter $\beta_{NPP}$ (% ppm$^{-1}$) and the fraction of denitrification ($f_{denit}$) simulated by the 10 Earth System Models (ESMs) in boreal regions (50°–90°N) during the period 1960–2014.** The black line and shaded area are the best-fit regression line and its 95% confidence interval, respectively, across the 10 ESMs. The unfilled symbols indicate parameter $\beta_{NPP}$ estimated from the 1pctCO2-bgc experiments, with the changes in the $CO_2$ concentration equivalent to that during the period 1960–2014. The unfilled symbols were used only for comparison and not for showing the negative correlation with $f_{denit}$, due to the limited number of ESMs (n = 5) for which the 1pctCO2-bgc experiment data was available. Source data are provided in Source Data file.

may also propagate to future scenarios, and, in turn, exaggerate the future N limitation on the C sink.

In summary, the isotope-benchmarking based $f_{denit}$ indicates that most CMIP6 ESMs overestimate $f_{denit}$, and shows little improvement over the CMIP5 models[8]. Due to the overestimation of $f_{denit}$ in the CMIP6 ESMs, the denitrification N loss is overestimated by 92%. These large overestimations of $f_{denit}$ and denitrification N loss suggest that the denitrification and/or related N cycle processes are underrepresented[4,17,48]. Furthermore, we found that this overestimation in denitrification N loss is closely related to the exaggeration of the N limitation on the simulation of plant growth under elevated $CO_2$ in the CMIP6 ESMs. Thus, to improve projections of the future land C sink, we call for improvements in the representation of denitrification processes, e.g., by incorporating the global distribution of microbial symbionts and its dynamics[22,23,45], and the changes in the soil oxygen condition (aerobiotic or anaerobic) and its heterogeneity[25]. Combining with recent advances[27], our isotope-benchmarking approach could be further used to partition the gaseous N loss into its components (e.g., $N_2O$, NO and $N_2$), allowing for a more refined assessment of ESMs. Overall, our upscaled global map of soil $\delta^{15}N$ provides a useful tool and a benchmark for constraining N-loss pathways in ESMs, highlighting that the representation of the N cycle needs to be improved in ESMs.

## Methods
### Global map of soil $\delta^{15}N$
To produce the global soil $\delta^{15}N$ map, we used a global soil $\delta^{15}N$ dataset comprising 5887 direct measurements (5609 measurements from Craine et al.[28] and 278 from Sena-Souza et al.[26]). As the original soil $\delta^{15}N$ dataset from Craine et al.[28] covers multiple soil depths and contains soil samples from various sites, we used only the $\delta^{15}N$ data from soils with depth ≤30 cm, while $\delta^{15}N$ data with the following conditions were excluded: (a) soil depth >30 cm; (b) C:N ratio is too low (< 1 gC gN$^{-1}$) to be considered as natural; (c) N concentration is too low (< 0.02 mg g$^{-1}$) to be considered as natural; (d) the sample is only collected from organic horizon without mineral layers, or C concentration is too high

(> 610 mg g$^{-1}$) to be considered as mineral; (e) the sample is collected from litter layer, the top layer of the soil column; (f) the sample site is adjacent to a marine ecosystem, which may involve a lot of N transformation processes in aquatic/coastal ecosystems (e.g., benthic N fixation, upwelling, burial, and phytoplankton uptake)[49–51] (g) the sample is from cropland; (h) the sample is from pastures, drystocks, dairy and industrial sites. The soil $\delta^{15}N$ within the depth of 30 cm were averaged weighted by soil N content if multiple depths were measured. We adopted 16 predictors with gridded fields: three climate drivers (precipitation (P), temperature (T) and aridity index–the ratio of precipitation over potential evapotranspiration (PET)), seven soil properties (bulk density (BD), soil pH, fractions of clay, silt and sand, organic carbon (OC), and soil C:N ratio (C/N)), three abundances of microbial symbionts (arbuscular mycorrhizal (AM), ectomycorrhizal (ECM) fungi, and N fixing bacteria (Nfix)), gross primary production (GPP), and $NH_x$ and $NO_y$ depositions (Supplementary Table 7). The 0.5° × 0.5° monthly P, T and PET data (1981–2018) were obtained from the Climatic Research Unit (CRU) Time-Series (TS) v4.03 datasets. The 10 km × 10 km BD, pH, fractions of clay, silt and sand, OC, and C/N were obtained from the Global Soil Datasets for Earth System Modelling produced by Beijing Normal University (BNU)[24], which provides soil information for eight soil layers (covering depths from 0 to 2.3 m); we used soil information for the upper four layers (-30 cm). The 1° × 1° natural abundance of AM, ECM, and Nfix were from Steidinger et al.[22] The 0.5° × 0.5° monthly GPP (1981–2016) and $NH_x$ and $NO_y$ depositions (2004–2015) were sourced from Keenan et al.[52] and Tian et al.[30], respectively. The 1° × 1° $NH_x$ and $NO_y$ depositions (2010) from the European Monitoring and Evaluation Programme (EMEP)[31] were also obtained, as an alternative to help account for the uncertainties in global soil $\delta^{15}N$ and $f_{denit}$ resulting from N deposition. Monthly data were averaged to obtain a mean annual value, and all these datasets were re-gridded to 0.1° × 0.1° spatial resolution.

First, we aggregated the 5887 site-level measurements of soil $\delta^{15}N$ into 933 0.1° × 0.1° grid cells (locations shown in Supplementary Fig. 1). Using the soil $\delta^{15}N$ and 16 predictors in the 933 grid cells, we employed a Random Forest (RF) algorithm to produce a global soil $\delta^{15}N$ map, using the well-established Python v3.8.5 package, RandomForestRegressor (Supplementary Text 1). This machine learning model explains 92% and 55% of the variances for training and testing samples, respectively (Supplementary Text 1; Supplementary Fig. 2). The K-fold (K = 10) cross-validation indicated that withholding 10% of the samples decreased the explained variances only slightly (Supplementary Fig. 3), i.e., the RF model is robust in predicting soil $\delta^{15}N$ across the globe. Compared to the linear models used by Amundson et al.[21], where climate drivers (T and P) were the only two predictors, the RF model increased the $R^2$ between observations and predictions across 933 grid cells from 0.20 to 0.93 and decreased the RMSE from 2.82‰ to 0.77‰ (Supplementary Fig. 5). Moreover, the RF model indicated that microbial symbionts (Nfix and ECM) and $NO_y$ deposition, in addition to climate (T and P/PET), play crucial roles in predicting soil $\delta^{15}N$ (Supplementary Figs. 17–21). The crucial roles of microbial symbionts result from that the N fixing bacteria assimilates atmospheric $N_2$ into soil with its $\delta^{15}N$ signal close to zero, and the plants associated with ECM and AM have different pathways of N uptake from soil, with the isotope fractionation higher for ECM than AM[33,34].

### Global map of isotope-benchmarking based $f_{denit}$
Following the isotope balance equations proposed by Houlton et al.[19], Houlton and Bai[20], and Bai and Houlton[53], the soil $\delta^{15}N$ is determined by $\delta^{15}N$ of N input and the isotopic fractionation involved in the denitrification, volatilization and leaching processes, i.e.,

$$\delta^{15}N_{soil} = \delta^{15}N_{input} + f_{denit}\varepsilon_{denit} + f_{leach}\varepsilon_{leach} + f_{vol}\varepsilon_{vol} \quad (1)$$

where $\delta^{15}N_{soil}$ and $\delta^{15}N_{input}$ are $\delta^{15}N$ signals of soil and input, respectively; $f_{denit}$, $f_{leach}$, and $f_{vol}$ are fractions of N losses from denitrification, leaching and volatilization, respectively ($f_{denit} + f_{leach} + f_{vol} = 1$); $\varepsilon_{denit}$, $\varepsilon_{leach}$ and $\varepsilon_{vol}$ are corresponding fractionation factors. Despite the volatilization of $NH_3$ occurring mainly in agricultural regions or high pH soils and accounting for only <5% of the total N loss flux in natural ecosystems[46,54,55], this small $NH_3$ flux could have a substantial effect on soil $\delta^{15}N$ due to its very high fractionation effects (29–35‰)[34,56]. Thus, we used four $NH_3$ volatilization scenarios to analyze the impact of $f_{vol}$ on $f_{denit}$ and the derived denitrification N losses. There is only a small change of $f_{denit}$ (< 0.03 or < 7%) under the four $NH_3$ volatilization scenarios (Supplementary Text 4; Supplementary Tables 8 and 9). Because of the limited impact of $f_{vol}$ on $f_{denit}$ and the large uncertainty in the assumed $NH_3$ volatilization as 1% or 5% of total N losses, as well as the $NH_3$ volatilization flux not being available in the CMIP6 ESM outputs, we ignore $f_{vol}$ here, i.e., $f_{denit} + f_{leach} \approx 1$. Following Eq. (1), the fraction of N loss from denitrification ($f_{denit}$) can be derived[20] as:

$$f_{denit} = \frac{\delta^{15}N_{soil} - \delta^{15}N_{input} - \varepsilon_{leach}}{\varepsilon_{denit} - \varepsilon_{leach}} \qquad (2)$$

The N inputs include atmospheric wet and dry N depositions, biological N fixation (BNF), and rock N weathering[29,30,53,57]. By considering N inputs from these three sources, $\delta^{15}N_{input}$ in each grid cell can be obtained as follows:

$$\delta^{15}N_{input} = \frac{I_{dep}\delta^{15}N_{dep} + I_{bnf}\delta^{15}N_{bnf} + I_{rock}\delta^{15}N_{rock}}{I_{dep} + I_{bnf} + I_{rock}} \qquad (3)$$

where $I_{dep}$, $I_{bnf}$, and $I_{rock}$ are the N input fluxes from deposition, BNF and rock weathering, respectively. $\delta^{15}N_{dep}$, $\delta^{15}N_{bnf}$, and $\delta^{15}N_{rock}$ are $\delta^{15}N$ signals in atmospherically deposited N, BNF and rock N weathering, respectively. The $\delta^{15}N$ in atmospherically deposited N is typically in the range of −3–3‰[19,58,59], and we adopted a central value of 0‰. The $\delta^{15}N$ of BNF was reported to be $-2 \pm 2.2$‰[34], and again we adopted the central value of −2‰. The $\delta^{15}N$ of rock N varies greatly across rock types (e.g., igneous, sedimentary and others; Supplementary Table 10)[33,60] and thus we produced a global rock $\delta^{15}N$ map based on the lithologic composition of the Earth's continental surfaces generated by Dürr et al.[61] and the $\delta^{15}N$ signals of different rock types as summarized by Holloway and Dahlgren[60] (Supplementary Fig. 22). On average, the global mean of rock $\delta^{15}N$ weighted by rock N flux is 4.02 ‰, and the lower and upper bounds of this rock $\delta^{15}N$ are 1.47‰ and 6.57‰, respectively. To assess the uncertainty due to N inputs, we produced the isotope-benchmarking based $f_{denit}$ with six global maps of $\delta^{15}N_{input}$ using one dataset of rock N weathering from Houlton et al.[29] (10 Tg Nyr$^{-1}$), two global maps of N deposition from Tian et al.[30] (39 Tg Nyr$^{-1}$) and EMEP[31] (42 Tg Nyr$^{-1}$), three global maps of BNF from Peng et al.[32] (46, 44 and 81 Tg Nyr$^{-1}$) (Supplementary Table 1; Supplementary Text 2). Moreover, the impacts of global $\delta^{15}N$ signals of rock N fluxes were assessed by adopting three levels (low, medium and high) of the $\delta^{15}N$ signal for a given rock type (Supplementary Table 2; Supplementary Text 3).

The most widely used values of the fractionation factors involved in the derivation of the global map of $f_{denit}$ are summarized in Supplementary Table 11. Hydrological leaching has been reported to have quite minor fractionation effects[15,19,20], and thus we adopted a value of zero for $\varepsilon_{leach}$. Denitrification involves a chain of multiple chemical processes and its fractionation has been reported to have large variations, i.e., 10–20‰ in natural soil communities[20] and 31–65‰ in pure incubation conditions in the laboratory[33,34]. As this study uses $\delta^{15}N$ observations in natural soil, we followed Houlton and Bai[20] and Houlton et al.[8], and selected a 13‰ isotope effect for $\varepsilon_{denit}$ in our analysis, leading to the derivation of a conservative estimate of $f_{denit}$ (Fig. 1b). As

the choice of $\varepsilon_{denit}$ is expected to have substantial impacts on the derived global map of $f_{denit}$, we assessed the sensitivity of $f_{denit}$ to $\varepsilon_{denit}$ by varying the isotope effect from 10‰ to 20‰. Furthermore, we also adopted two contrasting temperature-dependent scenarios for $\varepsilon_{denit}$ to derive the global map of $f_{denit}$ (Supplementary Text 3; Supplementary Fig. 10; Supplementary Table 2).

We used a Monte Carlo approach to evaluate the uncertainty in $f_{denit}$ for each $0.1° \times 0.1°$ grid cell. In each grid cell, the uncertainty of soil $\delta^{15}N$ was captured by ensembles from the RF model and all involved parameters were assumed to have Gaussian distributions. Specifically, the $\delta^{15}N_{input}$ SD was assumed to be 5% of the $\delta^{15}N$ mean, and the impact of this percentage was examined by carrying out a sensitivity analysis (Supplementary Fig. 23). Following Bai and Houlton[53] and Bai et al.[15], the uncertainties in $\varepsilon_{denit}$ and $\varepsilon_{leach}$ were controlled within 4‰ and 2‰ (i.e., SD = 1.02 and 0.51‰), respectively.

## Simulated $f_{denit}$ and N losses in the CMIP6 ESMs

We collected historical N losses from the gaseous/denitrification and leaching pathways and land N stocks of 15 ESMs with N related outputs from CMIP6 (https://esgf-node.llnl.gov/search/cmip6/). The details of the ESMs, including their spatial resolution, and the experiments and variants, are summarized in Supplementary Table 12. The ESMs can be divided into families, with the ESMs in the same family having similar patterns of $f_{denit}$. Therefore, for comparison with our isotope-benchmarking based $f_{denit}$ (Fig. 1b), we selected only one ESM from each of these families. Thus, eight out of the 15 ESMs were screened out, and Fig. 2 shows the global patterns of $f_{denit}$ for the eight ESM families. In the following analysis, the EC-Earth3-Veg-CC model was excluded due to the magnitudes of its N losses being in error, while the ACCESS-ESM1-5 was excluded because its N losses and inputs were much higher than those of the other ESMs. With the remaining 13 ESMs, we evaluated the 10-year (2005–2014) means of global denitrification and total N losses, and the N loss weighted global means of $f_{denit}$ simulated by the ESMs. Furthermore, we evaluated the terrestrial N sink as the mean annual increase of land nitrogen stocks, and derived the N input of the ESMs as the sum of the terrestrial N sink and the N losses from denitrification and leaching pathways. Notice that we excluded crop and pastural areas for all global maps of $f_{denit}$ and N losses (both those inferred from soil $\delta^{15}N$ and those from ESMs), following the land cover map of HYDE v3.2[62].

## Denitrification N loss derived from the isotope-benchmarking based $f_{denit}$

With the isotope-benchmarking based $f_{denit}$, we first estimated the denitrification N losses at steady state as the products of our six sets of global maps of $f_{denit}$ and N inputs (Supplementary Table 1). Further, we used the isotope-benchmarking based $f_{denit}$ to re-allocate the total N losses simulated by the CMIP6 ESMs into denitrification and leaching pathways (Supplementary Table 6). Reallocating the total N losses with the isotope-benchmarking based $f_{denit}$ could result in some biases since the natural terrestrial ecosystems have been sequestering N in recent decades, while the isotope-benchmarking based $f_{denit}$ was derived under steady state conditions. Thus, we assessed the effects of terrestrial N sinks on the isotope-based $f_{denit}$ in Supplementary Text 6 (Supplementary Fig. 12). Across the 13 CMIP6 ESMs, the terrestrial N sink is $25 \pm 7$ Tg Nyr$^{-1}$ with its maximum and minimum values of 44 and 18 Tg Nyr$^{-1}$, respectively. Since a larger terrestrial N sink is expected to have a larger effect on isotope based $f_{denit}$, we selected the mean and maximum values of the N sink for this sensitivity analysis. Specifically, the mean terrestrial N sink (25 Tg Nyr$^{-1}$) could increase the soil $\delta^{15}N$ by 0.02‰, resulting in a 0.002 increase in the isotope-based $f_{denit}$. The maximum terrestrial N sink (44 Tg Nyr$^{-1}$) could increase the soil $\delta^{15}N$ by 0.04‰, which results in a 0.004 increase in the isotope-based $f_{denit}$. Overall, the terrestrial N sinks could, at most, result in a < 1% (0.004/0.42 = 1%) bias in $f_{denit}$ between steady and non-steady states.

**Parameter for plant growth response to elevated $CO_2$, $\beta_{NPP}$**

To estimate the parameter $\beta_{NPP}$ from the framework tailored for this purpose (i.e., 1% $yr^{-1}$ increasing $CO_2$ experiments)[7], we obtained the simulation results from the CMIP6 fully coupled (1pctCO2) and only biogeochemically coupled (1pctCO2-bgc) experiments. However, data from these two experiments are only available for five ESMs (Supplementary Table 12), so we also used a regression method to estimate the parameter $\beta_{NPP}$ using Eq. (4) and (5)[63,64]. We used historical simulation outputs of net primary production (NPP), precipitation (P), and temperature (T) for 10 ESMs for which these data were available in CMIP6, and $CO_2$ concentration trajectories as specified in the protocols of the CMIP6 experiments[65]. We focused on the $\beta_{NPP}$ analysis in boreal regions because N limitation on $\beta_{NPP}$ is expected in these regions. To eliminate the collinearity effects across P, T, and $CO_2$, we first evaluated the sensitivities of NPP to P and T with detrended values using a multivariate linear regression method, i.e.,

$$NPP_{de} = \hat{\alpha}_T T_{de} + \hat{\alpha}_P P_{de} + \hat{\alpha}_{const} + \xi_{NPP} \qquad (4)$$

where $NPP_{de}$, $T_{de}$, and $P_{de}$ are detrended values of NPP, T, and P, respectively; $\hat{\alpha}_T$ and $\hat{\alpha}_P$ are the regressed sensitivities of NPP to T and P, respectively; $\hat{\alpha}_{const}$ is the regression constant, and $\xi_{NPP}$ is the regression error. Next, we estimated the residual of NPP from P and T by using the sensitivities $\hat{\alpha}_T$ and $\hat{\alpha}_P$ as $Residual = NPP - \hat{\alpha}_T T - \hat{\alpha}_P P - \hat{\alpha}_{const}$. Finally, the parameter $\beta_{NPP}$ was estimated by linear regression between the residual of NPP and $CO_2$ concentration, i.e.,

$$Residual = \hat{\beta}_{NPP} CO_2 + \hat{\alpha}'_{const} + \xi_{residual} \qquad (5)$$

where $\hat{\beta}_{NPP}$ is the regressed parameter quantifying the sensitivity of NPP to $CO_2$ concentration; $\hat{\alpha}'_{const}$ is the regression constant, and $\xi_{residual}$ is the regression error. The $\beta_{NPP}$ derived from this regression method is close to that obtained by using simulations from the 1% $yr^{-1}$ increasing $CO_2$ experiments across the five ESMs for which the required simulations were available (Fig. 4).

## Data availability

The site-level soil $\delta^{15}N$ measurements were obtained from Craine et al.[28] and Sena-Souza et al.[26] (https://esajournals.onlinelibrary.wiley.com/action/downloadSupplement?doi=10.1002%2Fecs2.3223&file=ecs23223-sup-0001-DataS1.zip). The climate data from the Climatic Research Unit (CRU) Time-Series (TS) v4.03 datasets are available at: https://catalogue.ceda.ac.uk/uuid/10d3e3640f004c578403419aac167d82. The soil properties from the Global Soil Dataset for use in Earth System Model (GSDE) produced by Beijing Normal University (BNU) are available at: http://globalchange.bnu.edu.cn/research/soilw. The global maps of abundance of microbial symbionts (arbuscular mycorrhizal (AM), ectomycorrhizal (ECM), and N fixing bacteria (N-fix)) from Steidinger et al.[22] are available at https://static-content.springer.com/esm/art%3A10.1038%2Fs41586-019-1128-0/MediaObjects/41586_2019_1128_MOESM4_ESM.zip. The global map of the gross primary production (GPP) is from Keenan et al.[52]. The global maps of nitrogen depositions (NHx and NOy) were obtained from Tian et al.[30] (https://data.isimip.org/), and the European Monitoring and Evaluation Programme (EMEP)[31] (https://thredds.met.no/thredds/catalog/data/EMEP/Articles_data/Schwede_etal_Ndep_2018/catalog.html). Three sets of global BNF maps simulated by the CSCA-CNP model (with methods A, B and C) from Peng et al.[32] were obtained by requesting the data from the corresponding authors. Rock weathering N flux from Dass et al.[57] is available at: https://datadryad.org/stash/dataset/doi:10.5061/dryad.5x69p8d1x. The global map of the lithologic composition of Earth's continental surfaces from Dürr et al.[61] was obtained by requesting the data from the corresponding author. All the historical simulation outputs of the ESMs are available from CMIP6 (https://esgf-node.llnl.gov/search/cmip6/). The global maps of soil $\delta^{15}N$, $f_{denit}$ and N loss produced in this study, as well as their uncertainty ranges, have been deposited at Figshare Database, and are publicly available (https://doi.org/10.6084/m9.figshare.22147283.v3)[66]. Source data for Fig. 3 and Fig.4 are provided with this paper. Source data are provided with this paper.

## Code availability

The python code for the Random Forest algorithm used to produce the global soil $\delta^{15}N$ map and the isotope mass balance model for deriving the fraction of denitrification N loss is available at: https://github.com/myFeng818/Codes-for-global-d15N-map-and-isotope-based-fdenit.git[67]. The MATLAB code used to regress the parameter for plant growth response to elevated $CO_2$ is available at: https://github.com/myFeng818/Codes-for-the-regression-of-Beta-and-NPP.git[68].

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

## Acknowledgements

The study was supported by the National Key Research and Development Program of China (2022YFF0801302), and the National Natural Science Foundation of China (grants 41722101, 41830643 and 52209003). P.C. acknowledges support from the 4C project funded by the European Commission Grant (agreement ID: 821003), and P.C. and D.S.G. acknowledge support from the CALIPSO project funded by the Schmidt's Futures Foundation. We are very grateful to the many scientists for their contribution to soil and vegetation δ¹⁵N observations, and for making these datasets available. We thank the climate modelling groups involved in CMIP5 and CMIP6 for producing and making available their model outputs.

## Author contributions

S.P. designed the study. M.F. coded the random forest model and nitrogen isotope model with substantial help from Y.W. M.F. performed the analysis with help of Y.W. and G.L., and M.F. created all the figures. P.C. and D.S.G. provided valuable comments on the analysis. S.P., M.F., G.L., and Y.X. wrote the first draft of the manuscript. P.C., D.S.G., Y.W., J.C., Y.F., B.Z.H., and Y.S. contributed to writing and commenting on the draft manuscript.

## Competing interests

The authors declare no competing interests.
