## [Peer Review File · Nature Communications]

Overestimated nitrogen loss from denitrification for natural terrestrial ecosystems in CMIP6 Earth System ModelsReviewer #1 (Remarks to the Author):

This paper is an important contribution to using ^{15}N mass balance in benchmarking estimates of terrestrial denitrification in ESMS and demonstrating that most current ESMS represent N losses poorly. The work builds on efforts that use first principles of isotope mass balance and the degree to which certain ecosystem process discriminate more than others against heavy N. The authors expand on these efforts by comparing isotope driven estimates with those produced by CMIP6 models that have a N cycle. I particularly appreciate the non-steady state simulations with the Beta factor and as a test of the robustness of soil ^{15}N approach. Of course, this sensitivity analysis could be done with many of the hypothesized drivers, but I don't think that is entirely necessary here. This paper relies heavily on machine learning algorithms to generate global maps of soil ^{15}N and couple these with globally gridded drivers including N inputs.

While I think the major conclusion of the paper—as emphasized in the title—that ESMS massively overestimate denitrification from land is well supported and important, I don't think one even needs ^{15}N to demonstrate this. A f_{denit} near 1 for much of the land surface violates observation and theory and simply does not make sense. Nevertheless, that's a major strength of this paper, bringing data analysis to bear to evaluate model assumptions and approximations, which is critical to better representation of the N cycle in ESMS.

Similar to recent work using ^{14}C to benchmark of soil C age in land models showing widespread overestimation of soil C sequestration, this paper concludes large overestimation of N losses in models. While this work is a step forward, I dislike how ^{15}N benchmarking is treated and discussed. All benchmarks have assumptions and error. However, unlike for example atmospheric CO_2 , inventories of emissions etc., isotope derived denitrification is not a direct observation, it is a model, as the authors are obviously well aware. It should not be treated as definitively in the language. In fact, theoretically, I'm not sure that the work can actually determine "overestimation" without direct gas loss constraints. For example "the actual value" of f_{denit} , line 185. I suggest something like the "isotope-based model estimate..." throughout. All fine as long as the approximation is explicit. The fundamentals of ecosystem ^{15}N mass balance simply are, but there is still much unknown about how ^{15}N imprints on bulk soils and the time scales involved.

Further, critique of the ways that EMS parameterize N losses, many of which are conceptually quite bizarre and detached from observation, is not new and it would be good to reference these works outside that by the authors themselves. Other approaches have also concluded that excessive and unrealistic N gas losses are unsustainable and could induce greater N limitation.

Consider discussing recent similar isotope mass balance analysis of $^{15}\text{NO}_3$ and soil ^{15}N by Harris et al. (Nature communications 2022) as another "observational" constraint. They use atmospheric N_2O measures and inversion as an actual benchmark (Not possible with N_2). They estimate 13.9Tg $\text{N}_2\text{O-N/yr}$ versus this studies 38-46 Tg N gas/year and assign a prominent forcing to anthropogenic N deposition. I suspect this paper came out after the present paper was submitted, but likely worth mentioning.

Please clarify how the Craine isotope data set was used. While the majority of samples are mineral soils <30 cm, the data set includes numerous values from organic soils, soils > 30 cm to 100cm depth. Further, careful inspection of the data reveals many samples from questionable "natural soils". While crops and urban soils were excluded, there are many agricultural land practices (e.g., pasture) in the data. It is important to clarify how these data were handled and how the assumptions may affect your conclusions.

It would be useful to have results of f_{denit} , N inputs and losses partitioned by major biomes. At least in a supplementary table or something. This should be a relatively straightforward aggregation.

L81: "N-input weighted"??

L95: Define upfront what is meant by "N input weighted" throughout. Why do this? Because it was one of the top three predictors in the machine learning analysis. Not clear.

L164: Important point about "atmospheric chemistry" but unclear. Clarify.

L169: "stream and river discharges" as large rivers already have already lost considerable N loads in transport

L185: "actual" value...

L271: This amp (S Fig 18) shows fdenit with different lithology 15N assumptions not rock 15N.

Supplementary L60-65: What is this saying?

Reviewer #2 (Remarks to the Author):

Summary: Feng et al. produce a new global map of soil 15N and use an isotope mass balance approach developed by Houlton et al. (2006, PNAS; Houlton & Bai 2009, PNAS) to estimate global terrestrial N losses from denitrification that amount to roughly half the magnitude (38 v 73 Tg N/yr) of those simulated by most of the recent set of Earth System Models used by the IPCC (CMIP6). This analysis expands on a similar study by Houlton et al. (2015, Nature Climate Change) that used a global soil d15N map extrapolated from a few dozen locations and compared with two versions of the land surface model in one ESM (CLM 4.0 and 4.5) to show that both versions greatly overestimated terrestrial N losses to denitrification and failed to capture global spatial patterns. The authors here take a similar approach with much more detailed soil 15N information derived from two datasets containing > 6000 measurements (in > 1000 grid cells). The improved global map yields a slightly lighter mean value (4.9 v 5.5 per mil). They also substantially expand the models examined to show that most of the ESMs used in the recent round of IPCC climate projections (CMIP6) overestimate denitrification, and they illustrate that models simulating greater relative loss of N to denitrification have more N limitation of productivity in boreal latitudes.

Review: The submission provides a robust update to the earlier related studies using isotope mass balance to infer global denitrification rates. Its most novel contributions are: 1) the new soil d15N map (Fig. 1), which could use some additional discussion noted below; 2) the comparison with a suite of CMIP6 models, showing that denitrification overestimates are widespread across models used in the recent IPCC assessment; and 3) the demonstration that overestimating denitrification N losses will exacerbate N limitation of carbon uptake by high-latitude ecosystems. It would be helpful if the manuscript clearly noted how its new global f-denit estimates (or total N gas fluxes, Tg N/yr) differed from prior uses of this approach. Together these are substantive contributions to the literature, and I support publication after revision to clarify items noted below.

The new soil map is one of the main new products of this research, developed with new a statistical model to fit and extend soil measurements compiled previously into two published datasets. As noted below, the Methods should clarify specifically how the soil d15N datasets were used to produce the new soil d15N maps here. The authors note that their new analysis uses not only temperature and precipitation (or here, aridity) as done previously, but also find important roles for microbial associations (with N fixers and ectomycorrhizae) and NOy deposition, but these relationships aren't shown anywhere. The Supplement (Fig. S-17) reports the relative importance of these

variables, but the relationships' themselves -- direction or magnitude – were not reported. These results are important to include for both their insights and to explain how this important map was produced.

Detailed comments:

Line 66. “We collected 6102 direct measurements of soil d15N from the literature... (ref. 24, 25)” Rephrase, such as replacing “collected” with “used” or similar to indicate that these measurements were collected by the two cited references rather than newly assembled from the original literature as part of this manuscript.

Line 67 and line 219. Soil d15N varies tremendously with soil depth, but depth is ambiguous here. Clarify and provide detail how the mapped soil d15N values were produced. What soil depth is mapped and used in denitrification calculations? How were the field measurements of 15N from various soil depths treated or combined? The Methods (line 219) mention using information for other soil properties “for the upper four layers (~30 cm),” and the main soil d15N dataset used here (Craine et al., 2015, Sci. Reports) contained measurements “from surface (< 30 cm) mineral soil” spanning various depths and excluding organic horizons. Because the map of soil d15N map produced here (Fig. 1a) is both a primary new product and an essential part of the denitrification calculations (Fig. 1b), some additional explanation is needed on how it was generated.

Line 78-79 + 239-249: The text states that the new global soil d15N map “performs well in capturing the nonlinear relationships between d15N observations and predictors” and reports an overall fit, but does not report the various nonlinear relationships in either the main text or Supplement. Please show the relationships with the most important predictors, and provide corresponding information on their fits at least in the Supplement if not the main text. It could be interesting to also compare with relationships reported by Craine et al., (2015), who assembled the main soil d15N dataset used here, as well as with relationships reported by Amundson et al.,(2003) mentioned qualitatively in the Methods.

~**Line 97.** How does the globally averaged value of 42% for f-denit (denitrification / total N losses) reported here compare with prior analyses using the isotope mass balance approach? (ref. 8, 18) How do the overall estimates of denitrification (Tg N/yr) compare?

Line 114-117. Continued problems with denitrification in the current version of CESM have also been reported elsewhere (e.g., Nevison et al., 2022, Ecological Applications DOI: 10.1002/eap.2528 and related).

Methods:

Line 258. Is Equation 2 correct? Rearranging Equation 1 to solve for f-denit should put d15N-input in the numerator instead of d15N-leach (as shown).

Line 263. It would be helpful to include in this section values used for the N input fluxes (Tg N/yr) and corresponding d15N values for all three of the N input terms (deposition, biological fixation, rock weathering). All terms do appear in the Supplement and several do appear in the Methods; adding brief specification of weighted mean rock d15N and the N input fluxes to the main text would save readers chasing down these important values from the Supplementary documents.

Responses to Reviewers

To Reviewer #1:

[Reviewer #1 General Comments]

[Reviewer #1 General Comment 1]

1. *This paper is an important contribution to using ^{15}N mass balance in benchmarking estimates of terrestrial denitrification in ESMs and demonstrating that most current ESMs represent N losses poorly. The work builds on efforts that use first principles of isotope mass balance and the degree to which certain ecosystem process discriminate more than others against heavy N. The authors expand on these efforts by comparing isotope driven estimates with those produced by CMIP6 models that have a N cycle. I particularly appreciate the non-steady state simulations with the Beta factor and as a test of the robustness of soil ^{15}N approach. Of course, this sensitivity analysis could be done with many of the hypothesized drivers, but I don't think that is entirely necessary here. This paper relies heavily on machine learning algorithms to generate global maps of soil ^{15}N and couple these with globally gridded drivers including N inputs.*

[Response] We thank the reviewer for the positive evaluations and valuable comments/suggestions on our manuscript. The detailed point-by-point responses are listed following each comment/suggestion.

[Reviewer #1 General Comment 2]

2. *While I think the major conclusion of the paper—as emphasized in the title—that ESMs massively overestimate denitrification from land is well supported and important, I don't think one even needs ^{15}N to demonstrate this. A f_{denit} near 1 for much of the land surface violates observation and theory and simply does not make sense. Nevertheless, that's a major strength of this paper, bringing data analysis to bear to evaluate model assumptions and approximations, which is critical to better representation of the N cycle in ESMs.*

[Response] We thank the reviewer for the positive feedbacks.

[Reviewer #1 General Comment 3]

3. Similar to recent work using ^{14}C to benchmark of soil C age in land models showing widespread overestimation of soil C sequestration, this paper concludes large overestimation of N losses in models. While this work is a step forward, I dislike how ^{15}N benchmarking is treated and discussed. All benchmarks have assumptions and error. However, unlike for example atmospheric CO_2 , inventories of emissions etc., isotope derived denitrification is not a direct observation, it is a model, as the authors are obviously well aware. It should not be treated as definitively in the language. In fact, theoretically, I'm not sure that the work can actually determine "overestimation" without direct gas loss constraints. For example "the actual value" of f_{denit} , line 185. I suggest something like the "isotope-based model estimate..." throughout. All fine as long as the approximation is explicit. The fundamentals of ecosystem ^{15}N mass balance simply are, but there is still much unknown about how ^{15}N imprints on bulk soils and the time scales involved.

[Response] We agree that all benchmarks have assumptions and errors and that the isotope-based estimates should not be taken as direct observations. First, our isotope mass balance model is based on the steady state assumption, and the impacts of this assumption on the isotope based f_{denit} were tested to be minor (Supplementary Text S6). Second, the isotope mass balance model is sensitive to the isotope fractionation of denitrification, whose effects were tested carefully in Supplementary Text S3. Despite above two points, we have to admit that our derived f_{denit} have some uncertainty and should not be taken as observations. Therefore, in the revised manuscript, we avoided using "actual value", and adopted the terms of "value", "denitrification N loss", "isotopically constrained estimate" or "model estimates" instead. In Line 197 (or Line 185 in previous manuscript), the "the actual value of f_{denit} " has been modified as "the value of f_{denit} "; in Line 169, the "actual denitrification N loss" has been modified as "denitrification N loss".

Moreover, we also added a sentence in Lines 177–180 to clarify the definition of model bias as follows:

"Note that the model bias is defined as the difference between isotopically constrained estimates and ESMs' simulated values, which provides an approximation of the true model bias as we lack direct observations of N losses at global scale."

[Reviewer #1 General Comment 4]

4. Further, critique of the ways that ESM parameterize N losses, many of which are conceptually quite bizarre and detached from observation, is not new and it would be good to reference these works outside that by the authors themselves. Other approaches have also concluded that excessive and unrealistic N gas losses are unsustainable and could induce greater N limitation.

[Response] Thanks for this nice suggestion. We have added a sentence in Lines 120–121 to indicate that the case of overestimated f_{denit} in our study is consistent with those in Nevison *et al.* (2022) and Thomas *et al.* (2013):

“The overestimation of f_{denit} is in line with the previous isotope based analysis⁸ and observation based comparisons^{9–10}.”

We have also added these two references of Nevison *et al.* (2022) and Thomas *et al.* (2013) to indicate the overestimation of denitrification in Lines 122–124:

“Moreover, the overestimated f_{denit} is also found in CESM^{8–10}, one out of the two CMIP5 ESMs that included nitrogen-carbon interactions, suggesting little improvement has been made in the representation of denitrification in this ESM.”

9. Nevison, C., *et al.* Nitrification, denitrification, and competition for soil N: Evaluation of two Earth System Models against observations. *Ecol. Appl.* **32**, e2528 (2022).

10. Thomas, R. *et al.* Global patterns of nitrogen limitation: confronting two global biogeochemical models with observations. *Glob. Change Biol.*, **19**: 2986–2998 (2013)..

[Reviewer #1 General Comment 5]

5. Consider discussing recent similar isotope mass balance analysis of $^{15}\text{NO}_3$ and soil ^{15}N by Harris *et al.* (Nature communications 2022) as another “observational” constraint. They use atmospheric N_2O measures and inversion as an actual benchmark (Not possible with N_2). They estimate 13.9Tg N_2O -N/yr versus this study 38–46 Tg N gas/year and assign a prominent forcing to anthropogenic N deposition. I suspect this paper came out after the present paper was submitted, but likely worth mentioning.

[Response] Thanks for this nice paper. Indeed, we saw this paper after our paper was submitted. In Harris *et al.* (2022), the gaseous N emissions from nitrification (NO and N₂O) and denitrification (N₂O and N₂) were 25.2±1.4 and 21.2±1.7 Tg N yr⁻¹, respectively. The total global gaseous N emissions were estimated to be 46±2 Tg N yr⁻¹ (including croplands) in Harris *et al.* (2022). As the gaseous N emissions were ONLY estimated from natural terrestrial ecosystems in this study (45±9 Tg N yr⁻¹), our estimate could be larger than Harris *et al.* (2022) for natural terrestrial ecosystems. Moreover, the denitrification N loss was also estimated by Houlton and Bai (2009) (44 Tg N yr⁻¹) and Bai *et al.* (2012) (47±8 Tg N yr⁻¹). Thus, we compared the estimates of total gaseous N losses in this study with the cases in Harris *et al.* (2022), Bai *et al.* (2012), and Houlton and Bai (2009) in Line 158–163:

“By applying the isotope-benchmarking based f_{denit} under the steady state assumption (total N losses are equal to total N inputs), we estimated the denitrification N loss from global natural terrestrial ecosystems as 45±9 Tg N yr⁻¹ (Fig. 3; Supplementary Table 1; Supplementary Table 4; Supplementary Fig. 15) using six sets of global N inputs (107±19 Tg N yr⁻¹), close to the recent estimates of 44–47 Tg N yr⁻¹ using similar isotope based framework but with different parameters and global N inputs^{15,20}.”

We also added the potential application of isotope based f_{denit} to “partitioning gaseous N loss into its components (e.g., N₂O, NO, and N₂)” in the implication as follows (Line 214–216):

“Combining with recent advances²⁷, our isotope-benchmarking approach could be further used to partition the gaseous N loss into its components (e.g., N₂O, NO and N₂), allowing for a more refined assessment of ESMs.”

27. Harris, *et al.* Warming and redistribution of nitrogen inputs drive an increase in terrestrial nitrous oxide emission factor. *Nat. Commun.* **13**, 4310 (2022).

[Reviewer #1 General Comment 6]

6. Please clarify how the Craine isotope data set was used. While the majority of samples are mineral soils <30 cm, the data set includes numerous values from organic soils, soils > 30 cm to

100cm depth. Further, careful inspection of the data reveals many samples from questionable “natural soils”. While crops and urban soils were excluded, there are many agricultural land practices (e.g., pasture) in the data. It is important to clarify how these data were handled and how the assumptions may affect your conclusions.

[Response] Thanks for this valuable suggestion. In the previous manuscript, we followed the data processing method in Craine *et al.* (2015), and used only the soil $\delta^{15}\text{N}$ data from mineral soils with depth ≤ 30 cm, while the soil $\delta^{15}\text{N}$ data with the following conditions were excluded:

- (a) Soil depth > 30 cm;
- (b) Low C:N ratio: C:N ratio is too low to be considered as natural;
- (c) Low N concentration: N concentration is too low to be considered as natural;
- (d) Organic: C concentration is too high to be considered as mineral;
- (e) Marine: the sample site is adjacent to a marine ecosystem;
- (f) Litter: the sample is collected from litter layer, and should not be considered as mineral soil;
- (g) Crops: soils are from cropland.

In the revised manuscript, we further excluded the soil $\delta^{15}\text{N}$ collected from pastures, drystocks, dairy and industrial sites ($n=197$, $197/(5806+278)=3\%$ of original data points), and obtained 5887 soil $\delta^{15}\text{N}$ data points from natural terrestrial ecosystems. Compared to the cases in the previous manuscript, we found minor changes in the estimates of isotope-benchmarking based f_{denit} (<0.01) and denitrification N losses (<1 Tg N yr⁻¹). We have updated all the calculations in the text, figures and tables in the revised manuscript and supplementary materials.

In the Methods (Line 223–230), we have added the description of data processing method of soil $\delta^{15}\text{N}$ datasets from Craine *et al.* (2015) as follows:

“As the original soil $\delta^{15}\text{N}$ datasets from Craine *et al.*²⁸ cover multiple soil depths and contain soil samples from various sites, we used only the $\delta^{15}\text{N}$ data from mineral soils with depth ≤ 30 cm, while $\delta^{15}\text{N}$ data with the following conditions were excluded: (a) soil depth > 30 cm; (b) C:N ratio is too low to be considered as natural; (c) N concentration is too low to be

considered as natural; (d) the C concentration is too high to be considered as mineral; (e) the sample site is adjacent to a marine ecosystem; (f) the sample is collected from litter layer; (g) the sample is from cropland; (h) the sample is from pastures, drystocks, dairy and industrial sites.”

[Reviewer #1 General Comment 7]

7. It would be useful to have results of f_{denit} , N inputs and losses partitioned by major biomes. At least in a supplementary table or something. This should be a relatively straightforward aggregation.

[Response] Thanks for this valuable suggestion. Using a land cover map of HYDE v3.2, we have evaluated the biome-level estimates of N inputs, f_{denit} and denitrification N losses, which were summarized in Table R1 (also presented as Supplementary Table 4):

Table R1 (also presented as Supplementary Table 4). The biome-level estimates of total N inputs ($Tg\ N\ yr^{-1}$), f_{denit} (-) and denitrification N losses ($Tg\ N\ yr^{-1}$) with different combinations of N deposition (Tian et al.³ and EMEP⁴) and biological nitrogen fixation (Method A, B, and C from Peng et al.⁸).

	TrBE	TrBR	TeNE	TeBE	TeBS	BrNE	BrBS	BrNS	C3 grass	C4 grass	Total
Total N Inputs ($Tg\ N\ yr^{-1}$)											
Tian+A	18.0	9.9	7.3	7.0	8.6	4.4	4.4	1.7	19.0	14.1	94.4
Tian+B	18.1	9.9	7.2	6.8	8.4	4.1	4.2	1.7	17.9	14.1	92.4
Tian+C	25.3	12.2	8.4	8.6	9.9	8.2	7.3	3.8	28.0	18.0	129.7
EMEP+A	19.8	10.5	8.2	8.5	9.5	3.2	3.6	1.0	17.4	16.0	97.6
EMEP+B	19.2	10.3	8.1	8.4	9.4	3.2	3.5	1.0	16.7	15.8	95.6
EMEP+C	26.5	12.6	9.3	10.2	10.8	7.3	6.7	3.1	26.8	19.7	132.9
Mean	21.2	10.9	8.1	8.3	9.4	5.1	4.9	2.0	21.0	16.3	107.1
f_{denit} (-)											
Tian+A	0.51	0.53	0.37	0.32	0.37	0.26	0.29	0.23	0.33	0.48	0.41
Tian+B	0.51	0.53	0.37	0.32	0.37	0.25	0.28	0.23	0.33	0.48	0.41
Tian+C	0.55	0.55	0.39	0.34	0.38	0.32	0.34	0.31	0.37	0.50	0.43
EMEP+A	0.51	0.52	0.34	0.29	0.34	0.27	0.29	0.23	0.32	0.46	0.41
EMEP+B	0.52	0.53	0.35	0.30	0.35	0.26	0.29	0.22	0.33	0.46	0.41
EMEP+C	0.55	0.55	0.37	0.32	0.36	0.34	0.35	0.33	0.37	0.49	0.43

Mean	0.53	0.53	0.37	0.32	0.36	0.28	0.31	0.26	0.34	0.48	0.42
	Denitrification N losses (Tg N yr ⁻¹)										
Tian+A	9.2	5.2	2.7	2.3	3.1	1.1	1.3	0.4	6.4	6.7	38.4
Tian+B	9.3	5.2	2.7	2.2	3.1	1.0	1.2	0.4	5.9	6.7	37.7
Tian+C	13.8	6.7	3.3	2.9	3.7	2.6	2.5	1.2	10.3	9.0	56.0
EMEP+A	10.0	5.4	2.9	2.5	3.3	0.9	1.1	0.2	6.0	7.3	39.7
EMEP+B	10.0	5.5	2.8	2.5	3.2	0.8	1.0	0.2	5.5	7.3	39.0
EMEP+C	14.6	6.9	3.5	3.2	3.9	2.5	2.3	1.0	10.0	9.6	57.4
Mean	11.2	5.8	3.0	2.6	3.4	1.5	1.6	0.6	7.3	7.8	44.7

[Reviewer #1 Detailed Comments]

[Reviewer #1 Detailed Comment 1]

1. L81: “N-input weighted”?? L95: Define upfront what is meant by “N input weighted” throughout. Why do this? Because it was one of the top three predictors in the machine learning analysis. Not clear.

[Response] The “N input weighted” indicates that a global average of soil $\delta^{15}\text{N}$ is obtained by using the grid level N inputs as weights. The reasons of adopting N inputs as weights are explained as follows:

The soil $\delta^{15}\text{N}$ should be averaged by using soil N content as weights, but we lack a reliable global map of soil N concentration here. Thus, we assumed that the soil N content is proportional to or linearly related to the N input (the product of N input flux and grid area) based on the steady state assumption. Considering that the grid land area was conventionally used as weights to obtain a global mean of soil $\delta^{15}\text{N}$, we have also checked the difference between “N-input weighted” and “area-weighted” soil $\delta^{15}\text{N}$, and found that these two weighted values are very close (<0.03‰).

Therefore, in Lines 82–84 of the revised manuscript, we have clarified the meaning of “N input weighted” mean, which is also presented as follows:

“The global soil $\delta^{15}\text{N}$ map has a mean of 4.8‰ weighted by grid level N inputs (proportional to soil N content at steady state, and estimated as the product of N input flux and grid area) (Fig. 1a; Supplementary Fig. 4), which is slightly lower than the previous estimate of 5.5‰^{8,21}.”

[Reviewer #1 Detailed Comment 2]

2. L164: Important point about “atmospheric chemistry” but unclear. Clarify.

[Response] We have added details to explain “atmospheric chemistry” as follows (Lines 172–174):
“This calculation suggests that the CMIP6 ESMs overestimate the denitrification N loss by 92%, which would further bias the atmospheric chemistry (e.g., atmosphere N₂O, NO, and NO₂ concentrations and attendant chemical processes) if resolved in the ESMs.”

[Reviewer #1 Detailed Comment 3]

3. L169: “stream and river discharges” as large rivers already have already lost considerable N loads in transport

[Response] Following the reviewer’s suggestion, we have replaced the term “river discharges” by “stream and river discharges”.

[Reviewer #1 Detailed Comment 4]

4. L185: “actual” value...

[Response] It has been removed.

[Reviewer #1 Detailed Comment 5]

5. L271: This amp (S Fig 18) shows f_{denit} with different lithology ¹⁵N assumptions not rock ¹⁵N.

[Response] Thanks for this comment. We would like to clarify that the Supplementary Fig. 22 (Supplementary Fig. 18 in the previous version) was produced from (1) the present-day **surficial** lithological map of the world from Dürr *et al.* (2005), and (2) $\delta^{15}\text{N}$ signals for different types of **rocks** collected from Holloway and Dahlgren (2002), summarized in Supplementary Table 10. Specifically, we produced the Supplementary Fig. 22 by assigning the rock-specific $\delta^{15}\text{N}$ signal to the surficial lithological map of the world.

To avoid the possible misunderstanding, we have corrected the title of Supplementary Fig. 22 (Supplementary Fig. 18 in the previous) as follows:

“Supplementary Fig. 22. Global maps of rock $\delta^{15}\text{N}$ signals. (a) is a medium level of rock $\delta^{15}\text{N}$; (b) and (c) are the lower and upper bounds of the rock $\delta^{15}\text{N}$, respectively. The global rock $\delta^{15}\text{N}$ maps were produced based on the surficial lithologic composition of Earth’s continental surfaces generated by Dürr et al.²⁶ and $\delta^{15}\text{N}$ signals of different rock types summarized in Supplementary Table 10.”

[Reviewer #1 Detailed Comment 6]

6. Supplementary L60-65: What is this saying?

[Response] We have modified this sentence as follows:

“Specifically, compared to Amundson *et al.*⁶, our global map has higher $\delta^{15}\text{N}$ signals in the Amazonia and lower $\delta^{15}\text{N}$ signals in Southeast and East Asia. Amundson *et al.*⁶ collected very limited soil $\delta^{15}\text{N}$ observations (~50) at 6 sites (4-6 elevations at each site), while we have utilized thousands of observations covering different climate zones, vegetation types and six continents. Thus, we believed that our map has higher confidence than the predicted map in Amundson *et al.*⁶. (Supplementary Fig. 1).”

References

Houlton, B. Z., Marklein, A. R. & Bai, E. Representation of nitrogen in climate change forecasts. *Nature Clim. Change*, **5**, 398-401 (2015).

Nevison, C., *et al.* Nitrification, denitrification, and competition for soil N: Evaluation of two Earth System Models against observations. *Ecol. Appl.* **32**, e2528 (2022).

Thomas, R. *et al.* Global patterns of nitrogen limitation: confronting two global biogeochemical models with observations. *Glob. Change Biol.*, **19**: 2986-2998 (2013).

Harris, *et al.* Warming and redistribution of nitrogen inputs drive an increase in terrestrial nitrous oxide emission factor. *Nat. Commun.* **13**, 4310 (2022).

Craine, J. M. *et al.* Ecological interpretations of nitrogen isotope ratios of terrestrial plants and soils. *Plant Soil*, **396**, 1-26 (2015).

Dürr, H. H. , Meybeck, M. & Dürr, S. H. Lithologic composition of the Earth's continental surfaces derived from a new digital map emphasizing riverine material transfer. *Global Biogeochemical Cycles*, **19**, GB4S10 (2005).

Holloway, J. M. & Dahlgren, R. A. Nitrogen in rock: Occurrences and biogeochemical implications. *Global Biogeochem. Cycles*, **16**, 1118 (2002).

To Reviewer #2:

[Reviewer #2 General Comments]

[Reviewer #2 General Comment 1]

1. Summary: Feng et al. produce a new global map of soil ^{15}N and use an isotope mass balance approach developed by Houlton et al. (2006, PNAS; Houlton & Bai 2009, PNAS) to estimate global terrestrial N losses from denitrification that amount to roughly half the magnitude (38 v 73 Tg N/yr) of those simulated by most of the recent set of Earth System Models used by the IPCC (CMIP6). This analysis expands on a similar study by Houlton et al. (2015, Nature Climate Change) that used a global soil $\delta^{15}\text{N}$ map extrapolated from a few dozen locations and compared with two versions of the land surface model in one ESM (CLM 4.0 and 4.5) to show that both versions greatly overestimated terrestrial N losses to denitrification and failed to capture global spatial patterns. The authors here take a similar approach with much more detailed soil ^{15}N information derived from two datasets containing > 6000 measurements (in > 1000 grid cells). The improved global map yields a slightly lighter mean value (4.9 v 5.5 per mil). They also substantially expand the models examined to show that most of the ESMs used in the recent round of IPCC climate projections (CMIP6) overestimate denitrification, and they illustrate that models simulating greater relative loss of N to denitrification have more N limitation of productivity in boreal latitudes.

[Response] We thank the reviewer for the positive evaluations and valuable comments/suggestions on our manuscript. The point-by-point responses are listed following each comment/suggestion.

[Reviewer #2 General Comment 2]

2. Review: The submission provides a robust update to the earlier related studies using isotope mass balance to infer global denitrification rates. Its most novel contributions are: 1) the new soil $\delta^{15}\text{N}$ map (Fig. 1), which could use some additional discussion noted below; 2) the comparison with a suite of CMIP6 models, showing that denitrification overestimates are widespread across models used in the recent IPCC assessment; and 3) the demonstration that overestimating denitrification N losses will exacerbate N limitation of carbon uptake by high-latitude ecosystems. It would be helpful if the manuscript clearly noted how its new global f_{denit} estimates (or total N

gas fluxes, Tg N/yr) differed from prior uses of this approach. Together these are substantive contributions to the literature, and I support publication after revision to clarify items noted below.

[Response] We thank the reviewer for these positive feedbacks and constructive suggestions. We agree that our manuscript could be improved by comparing our new global estimates of f_{denit} and total gaseous N losses with those from prior estimates. Following the reviewer's suggestion, we have (1) compared our isotope-based estimates of f_{denit} with those in Houlton *et al.* (2006), Houlton and Bai (2009), and Houlton *et al.* (2015), and added “... **slightly higher than previous area-weighted estimates of 26–40% under similar isotope-based framework but with a global map of soil $\delta^{15}\text{N}$ from Amundson *et al.*²¹ and different parameterizations^{8,19,20}**” in Lines 101–103; (2) compared our isotope-based estimates of total gaseous N fluxes to those in Harris *et al.* (2022), Bai *et al.* (2012), and Houlton and Bai (2009), and added “... **close to the recent estimates of 44–47 Tg N yr⁻¹ using similar isotope based framework but with different parameters and global N inputs^{15,20}**” in Lines 161–163.

[Reviewer #2 General Comment 3]

3. *The new soil map is one of the main new products of this research, developed with new a statistical model to fit and extend soil measurements compiled previously into two published datasets. As noted below, the Methods should clarify specifically how the soil $\delta^{15}\text{N}$ datasets were used to produce the new soil $\delta^{15}\text{N}$ maps here. The authors note that their new analysis uses not only temperature and precipitation (or here, aridity) as done previously, but also find important roles for microbial associations (with N fixers and ectomycorrhizae) and NO_y deposition, but these relationships aren't shown anywhere. The Supplement (Fig. S-17) reports the relative importance of these variables, but the relationships' themselves -- direction or magnitude – were not reported. These results are important to include for both their insights and to explain how this important map was produced.*

[Response] We thank the reviewer for this valuable comment. Following the reviewer's suggestion, we have 1) added the data processing method in Methods (Line 223–230), and 2) added the linear regression relationships between soil $\delta^{15}\text{N}$ observations and six leading predictors identified by Random Forest (RF) model (Fig. R1, also added as Supplementary Fig. 18), and the nonlinear

partial dependence relationship between RF-predicted soil $\delta^{15}\text{N}$ and these six predictors (Fig. R2, also added as Supplementary Fig. 19).

(1) As replied in the *Reviewer #1* General comment 6, we added the description of data processing method in Methods (Line 223–230) as follows:

“As the original soil $\delta^{15}\text{N}$ datasets from Craine *et al.*²⁸ cover multiple soil depths and contain soil samples from various sites, we used only the $\delta^{15}\text{N}$ data from mineral soils with depth ≤ 30 cm, while $\delta^{15}\text{N}$ data with the following conditions were excluded: (a) soil depth > 30 cm; (b) C:N ratio is too low to be considered as natural; (c) N concentration is too low to be considered as natural; (d) the C concentration is too high to be considered as mineral; € the sample site is adjacent to a marine ecosystem; (f) the sample is collected from litter layer; (g) the sample is from cropland; (h) the sample is from pastures, drystocks, dairy and industrial sites.”

(2) We illustrated the linear regression relationships between soil $\delta^{15}\text{N}$ observations and six leading predictors in Fig. R1 (also added as Supplementary Fig. 18), and the nonlinear partial dependence relationships between RF-predicted soil $\delta^{15}\text{N}$ and these six predictors in Fig. R2 (also added as Supplementary Fig. 19). Figs. R1 and R2 highlight the crucial roles of microsymbionts (Nfixer, ECM, and AM) and N deposition, additional to the climate drivers, in predicting soil global soil $\delta^{15}\text{N}$, which are potential variables for improving model representations of the N cycle considering the N isotope benchmark.

Fig. R1. (also shown as Supplementary Fig. 18) The linear regression relationships between soil $\delta^{15}\text{N}$ observations and six leading predictors. The leading predictors are identified by random forest model: (a) N fixing bacteria (Nfixer), (b) Temperature (T), (c) Ectomycorrhizal fungi (ECM), (d) aridity index–precipitation over potential evapotranspiration (P/PET), (e) N deposition in form of NOy, (f) Arbuscular mycorrhizal fungi (AM).

Fig. R2. (also shown as Supplementary Fig. 19) The partial dependence plots between the RF-predicted $\delta^{15}\text{N}$ and six leading predictors. The partial dependence between RF-predicted $\delta^{15}\text{N}$ and (a) N fixing bacteria (Nfixer), (b) Temperature (T), (c) Ectomycorrhizal fungi (ECM), (d) aridity index–precipitation over potential evapotranspiration (P/PET), (e) N deposition in form of NOy, (f) Arbuscular mycorrhizal fungi (AM). The partial dependence was quantified by a well-established python package, `partial_dependence`, and plotted by the MATLAB software.

[Reviewer #2 Detailed comments]

[Reviewer #2 Detailed Comment 1]

1. Line 66. “We collected 6102 direct measurements of soil $\delta^{15}\text{N}$ from the literature... (ref. 24, 25)”
Rephrase, such as replacing “collected” with “used” or similar to indicate that these measurements were collected by the two cited references rather than newly assembled from the original literature as part of this manuscript.

[Response] It has been corrected as “used”.

[Reviewer #2 Detailed Comment 2]

2. Line 67 and line 219. Soil $\delta^{15}\text{N}$ varies tremendously with soil depth, but depth is ambiguous here. Clarify and provide detail how the mapped soil $\delta^{15}\text{N}$ values were produced. What soil depth is mapped and used in denitrification calculations? How were the field measurements of ^{15}N from various soil depths treated or combined? The Methods (line 219) mention using information for other soil properties “for the upper four layers (~30 cm),” and the main soil $\delta^{15}\text{N}$ dataset used here (Craine *et al.*, 2015, *Sci. Reports*) contained measurements “from surface (< 30 cm) mineral soil” spanning various depths and excluding organic horizons. Because the map of soil $\delta^{15}\text{N}$ map produced here (Fig. 1a) is both a primary new product and an essential part of the denitrification calculations (Fig. 1b), some additional explanation is needed on how it was generated.

[Response] We thank the reviewer for the valuable suggestions. Following the reviewer’s suggestions, we have added the description of how to use soil $\delta^{15}\text{N}$ datasets from Craine *et al.* (2015) in Methods (Line 223–230) as follows:

“As the original soil $\delta^{15}\text{N}$ datasets from Craine *et al.*²⁸ cover multiple soil depths and contain soil samples from different sites, we used only the $\delta^{15}\text{N}$ data from mineral soils with depth \leq 30 cm, while $\delta^{15}\text{N}$ data with the following conditions were excluded: (a) soil depth > 30 cm; (b) C:N ratio is too low to be considered as natural; (c) N concentration is too low to be considered as natural; (d) the C concentration is too high to be considered as organic; (e) the sample site is adjacent to a marine ecosystem; (f) the sample is collected from litter layer; (g)

the sample is from cropland; (h) the sample is from pastures, drystocks, dairy and industrial sites.”

[Reviewer #2 Detailed Comment 3]

3. Line 78-79 + 239-249: The text states that the new global soil $\delta^{15}\text{N}$ map “performs well in capturing the nonlinear relationships between $\delta^{15}\text{N}$ observations and predictors” and reports an overall fit, but does not report the various nonlinear relationships in either the main text or Supplement. Please show the relationships with the most important predictors, and provide corresponding information on their fits at least in the Supplement if not the main text. It could be interesting to also compare with relationships reported by Craine *et al.* (2015), who assembled the main soil $\delta^{15}\text{N}$ dataset used here, as well as with relationships reported by Amundson *et al.* (2003) mentioned qualitatively in the Methods.

[Response] Thank the reviewer for this valuable comment. The random forest (RF) model indicated six leading predictors for soil $\delta^{15}\text{N}$ as N fixing bacteria (Nfixer), Temperature (T), Ectomycorrhizal fungi (ECM), aridity index–precipitation over potential evapotranspiration (P/PET), N deposition in form of NO_y (NO_y), and Arbuscular mycorrhizal fungi (AM). Thus, we have added Figs. R1 and R2, (shown in the response to General Comment 3) to illustrate the linear regression relationships between the soil $\delta^{15}\text{N}$ observations and these six leading predictors, as well as the nonlinear partial dependence relationships between RF-predicted soil $\delta^{15}\text{N}$ and these six leading predictors.

Following the reviewer’s suggestion, we compared the fitted relationships between soil $\delta^{15}\text{N}$ and mean annual temperature (MAT) and precipitation (MAP) in this study with the cases in Craine *et al.* (2015) (Fig. R3–R4; Supplementary Figs. 20–21 in the revised Supplementary). In both this study and Craine *et al.* (2015), we found similar positive linear relationships between soil $\delta^{15}\text{N}$ and MAT (Fig. R3). Moreover, by adopting a threshold of 8.5°C, we found similar piecewise linear relationships between soil $\delta^{15}\text{N}$ and MAT in both cases, i.e., soil $\delta^{15}\text{N}$ keeps almost the same when MAT < 8.5°C while increases with the increase of MAT (Fig. R4). However, soil $\delta^{15}\text{N}$ has weak relationship with MAP both either in linear or log schemes (Figs. R3 and R4).

Furthermore, we also compared the fitted relationships between soil $\delta^{15}\text{N}$ and MAT and MAP with the cases in Amundson *et al.* (2003). We adopted a multivariate linear regression method to fit the soil $\delta^{15}\text{N}$ to MAT and MAP, and obtained a regression function of $\delta^{15}\text{N} = 0.1287 \text{ MAT} - 0.0007 \text{ MAP} + 3.7217$. In Amundson *et al.* (2003), the fitted relationships between soil $\delta^{15}\text{N}$ and MAT and MAP is $\delta^{15}\text{N} = 0.1680 \text{ MAT} - 0.007 \text{ MAP} + 3.8864$ (with a soil depth ≤ 50 cm). This relationship is similar to the regression in this study, i.e., soil $\delta^{15}\text{N}$ is positively related to MAT while the relationship between soil $\delta^{15}\text{N}$ and MAP is quite weak.

In summary, the previous linear regression models with ONLY climate drivers have limited capability in predicting global soil $\delta^{15}\text{N}$. By comparing the RF model and previous linear regression models, we highlight the powerful ability of the RF model in capturing the nonlinear relationships between soil $\delta^{15}\text{N}$ and its predictors, and also substantial contributions of N fixing bacteria (Nfixer), ECM and AM in predicting global soil $\delta^{15}\text{N}$.

Fig. R3. (also shown as Supplementary Fig. 20) Comparison of the fitted relationships between soil $\delta^{15}\text{N}$ observations and mean annual temperature (MAT) and precipitation (MAP) in this study with those in Craine *et al.* (2015). (a) and (b) are linear regression relationship between soil $\delta^{15}\text{N}$

observations and MAT and MAP, respectively, in this study. (c) and (d) are linear regression relationship between soil $\delta^{15}\text{N}$ observations and MAT and MAP, respectively, in Craine *et al.* (2015).

Fig. R4. (also shown as Supplementary Fig. 21) Comparison of the nonlinear relationships between soil $\delta^{15}\text{N}$ observations and mean annual temperature (MAT) precipitation (MAP) in this study with those in Craine *et al.* (2015). (a) and (c) are piecewise linear relationships between soil $\delta^{15}\text{N}$ observations and MAT (with a threshold of 8.5 °C), in this study and Craine *et al.* (2015), respectively. (b) and (d) are regression relationships between soil $\delta^{15}\text{N}$ observations and log MAP, in this study and Craine *et al.* (2015), respectively.

[Reviewer #2 Detailed Comment 4]

4. ~Line 97. How does the globally averaged value of 42% for f_{denit} (denitrification/total N losses) reported here compare with prior analyses using the isotope mass balance approach? (ref. 8, 18) How do the overall estimates of denitrification (Tg N/yr) compare?

[Response] We have added the comparison between our isotope-based f_{denit} estimates with previous analysis as follows (Line 98–103):

“The ensemble of global maps of isotope-based f_{denit} are similar, providing a global N input weighted average of f_{denit} equal to 0.42 ± 0.01 (mean \pm standard deviation (SD)) (Supplementary Table 1), i.e., $42 \pm 1\%$ of N losses in natural ecosystems occur via the denitrification pathway, slightly higher than previous area-weighted estimates of 26–40% under similar isotope-based framework but with a global map of soil $\delta^{15}\text{N}$ from Amundson *et al.*²¹ and different parameterizations^{8,19,20}.”

Similarly, we also added the comparison between our estimates of denitrification N losses with previous estimates (Line 158–163):

“With the isotope-benchmarking based f_{denit} and a steady state assumption (total N losses are equal to total N inputs), we estimated the denitrification N loss from global natural terrestrial ecosystems as 45 ± 9 Tg N yr⁻¹ (Fig. 3; Supplementary Table 1; Supplementary Table 4; Supplementary Fig. 15) using six sets of global N inputs (107 ± 19 Tg N yr⁻¹), close to the recent estimates of 44–47 Tg N yr⁻¹ using similar isotope based framework but with different parameters and global N inputs^{15,20}.”

[Reviewer #2 Detailed Comment 5]

5. Line 114-117. Continued problems with denitrification in the current version of CESM have also been reported elsewhere (e.g., Nevison et al., 2022, Ecological Applications DOI: 10.1002/eap.2528 and related).

[Response] We have added a sentence in Lines 120–121 to indicate that the case of overestimated f_{denit} in our study is consistent with those in Nevison *et al.* (2022) and Thomas *et al.* (2013):

“The overestimation of f_{denit} is in line with the previous isotope based analysis⁸ and observation based comparisons^{9–10}.”

We have also added these two references of Nevison *et al.* (2022) and Thomas *et al.* (2013) to indicate the overestimation of denitrification in CESM (Lines 122–124):

“Moreover, the overestimated f_{denit} is also found in CESM⁸⁻¹⁰, one out of the two CMIP5 ESMs that included nitrogen-carbon interactions, suggesting little improvement has been made in the representation of denitrification in this ESM.”

9. Nevison, C., *et al.* Nitrification, denitrification, and competition for soil N: Evaluation of two Earth System Models against observations. *Ecol. Appl.* **32**, e2528 (2022).

10. Thomas, R. *et al.* Global patterns of nitrogen limitation: confronting two global biogeochemical models with observations. *Glob. Change Biol.*, **19**: 2986-2998 (2013)..

[Reviewer #2 Detailed Comment 6]

6. Line 258. Is Equation 2 correct? Rearranging Equation 1 to solve for f_{denit} should put $\delta^{15}\text{N}_{\text{input}}$ in the numerator instead of $\delta^{15}\text{N}_{\text{leach}}$ (as shown).

[Response] Sorry for this mistake. The Eq. (2) have been corrected by replacing $\delta^{15}\text{N}_{\text{leach}}$ with $\delta^{15}\text{N}_{\text{input}}$, with the corrected equation represented as follows:

$$f_{\text{denit}} = \frac{\delta^{15}\text{N}_{\text{soil}} - \delta^{15}\text{N}_{\text{input}} - \epsilon_{\text{leach}}}{\epsilon_{\text{denit}} - \epsilon_{\text{leach}}} \quad (2)$$

[Reviewer #2 Detailed Comment 7]

7. Line 263. It would be helpful to include in this section values used for the N input fluxes (Tg N/yr) and corresponding $\delta^{15}\text{N}$ values for all three of the N input terms (deposition, biological fixation, rock weathering). All terms do appear in the Supplement and several do appear in the Methods; adding brief specification of weighted mean rock $\delta^{15}\text{N}$ and the N input fluxes to the main text would save readers chasing down these important values from the Supplementary documents.

[Response] Thank the reviewer for this valuable suggestion. Following this suggestion, we have added three N input fluxes (deposition, biological fixation, rock weathering) in Methods (Lines 295–299) as follows:

“To assess the uncertainty due to N inputs, we produced the isotope-benchmarking based f_{denit} with six global maps of $\delta^{15}\text{N}_{\text{input}}$ using one dataset of rock N weathering from Houlton *et*

*al.*²⁹ (10 Tg N yr⁻¹), two global maps of N deposition from Tian *et al.*³⁰ (39 Tg N yr⁻¹) and EMEP³¹ (42 Tg N yr⁻¹), three global maps of BNF from Peng *et al.*³² (46, 44 and 81 Tg N yr⁻¹) (Supplementary Table 1; Supplementary Text 2).”

We also added the $\delta^{15}\text{N}$ signal for the three N input fluxes (deposition, biological fixation, rock weathering) in Methods (Lines 287–295) as follows:

“The $\delta^{15}\text{N}$ in atmospherically deposited N is typically in the range of -3–3‰^{19,55,56}, and we adopted a central value of 0‰. The $\delta^{15}\text{N}$ of BNF was reported to be $-2\pm 2.2\%$ ³⁴, and again we adopted the central value of -2‰. The $\delta^{15}\text{N}$ of rock N varies greatly across rock types (e.g., igneous, sedimentary and others; Supplementary Table 10)^{33,57} and thus we produced a global rock $\delta^{15}\text{N}$ map based on the lithologic composition of the Earth’s continental surfaces generated by Dürr *et al.*⁵⁸ and the $\delta^{15}\text{N}$ signals of different rock types as summarized by Holloway and Dahlgren⁵⁷ (Supplementary Fig. 21). On average, the global mean of rock $\delta^{15}\text{N}$ weighted by rock N flux is 4.02 ‰, and the lower and upper bounds of this rock $\delta^{15}\text{N}$ are 1.47‰ and 6.57‰, respectively.”

In the main, we added the brief specification of $\delta^{15}\text{N}$ and the N input fluxes in Lines 93–98 as follows:

“To account for the uncertainties in these N input data, we derived an ensemble of global maps of isotope-based f_{denit} using six sets of N inputs by combining one global map of rock N weathering²⁹ (10 Tg N yr⁻¹ with a global mean $\delta^{15}\text{N}$ of 4.02‰), two global maps of N deposition^{30,31} (an average of 40.7 Tg N yr⁻¹ with a constant $\delta^{15}\text{N}$ of 0‰) and three global maps of BNF³² (an average of 57 Tg N yr⁻¹ with a constant $\delta^{15}\text{N}$ of -2‰) (see Methods and Supplementary Text 2).”

References

Houlton, B. Z., Sigman, D. M. & Hedin, L. O. Isotopic evidence for large gaseous nitrogen losses from tropical rainforests. *Proc. Natl. Acad. Sci. U. S. A.*, **103**, 8745-8750 (2006).

Houlton, B. Z. & Bai, E. Imprint of denitrifying bacteria on the global terrestrial biosphere. *Proc. Natl. Acad. Sci. U. S. A.*, **106**, 21713-21716 (2009).

Houlton, B. Z., Marklein, A. R. & Bai, E. Representation of nitrogen in climate change forecasts. *Nature Clim. Change*, **5**, 398-401 (2015).

Harris, *et al.* Warming and redistribution of nitrogen inputs drive an increase in terrestrial nitrous oxide emission factor. *Nat. Commun.* **13**, 4310 (2022).

Nevison, C., *et al.* Nitrification, denitrification, and competition for soil N: Evaluation of two Earth System Models against observations. *Ecol. Appl.* **32**, e2528 (2022).

Thomas, R. *et al.* Global patterns of nitrogen limitation: confronting two global biogeochemical models with observations. *Glob. Change Biol.*, **19**: 2986-2998 (2013).

Craine, J. M. *et al.* Ecological interpretations of nitrogen isotope ratios of terrestrial plants and soils. *Plant Soil*, **396**, 1-26 (2015).

Amundson, R. *et al.* Global patterns of the isotopic composition of soil and plant nitrogen. *Global Biogeochem. Cycles*, **17**, 1031 (2003).

Reviewer #1 (Remarks to the Author):

The authors thought about my comments and addressed them well, including reevaluating data and conducting new analyses. The assumptions and approximations are more explicit and the paper is better. The one thing that I would like the authors to finally consider and provide is an explanation (can be brief statement or analysis) of the inference and mechanisms underlying the strong explanatory power of N fixer and EM abundance based on globally gridded data. The relative influence of these variables are strong in the models. Although the RF model should account for this, the global distribution of these symbioses are known to exhibit strong collinearity with Temp and other climate/latitudinal drivers. So what are the mechanisms? The description of these findings in results and supplemental are relatively muted, e.g. "Moreover, we would like to highlight the crucial roles of microsymbionts (Nfixer, ECM) and N deposition, which are potential variables for improving model..". Without asking too much detail, please provide some depth here as to why these tree-microbe associations should imprint on soil isotope based denitrification estimates.

Reviewer #2 (Remarks to the Author):

I was Reviewer 2 on the prior version of this manuscript. I have reviewed the revised manuscript and the authors' response to review. I concur with the comments by Reviewer 1, and I commend the authors on both the initial analyses and for the substantial work invested in reviewer questions.

The new Supplementary Figures 18 & 19 are fascinating, and provide important support for the main analysis. Although it would be nice for the main text to further discuss these findings on the direction and magnitude of key factors associated with soil d15N, I'm glad the figures are at least in the Supplement.

I also like the table added per Reviewer 1's request (Supplementary Table 4), showing biome-level estimates of N inputs and f-denit. Could an additional row be added reporting the area of each biome? That addition would allow conversion between units of total N flux (Tg N/yr) and flux per unit area.

Overall, the authors have nicely addressed my comments, although responses to one request from both reviewers raises even more questions: both Rev. 1 and I sought added explanation on how the authors used the Craine et al. soil 15N dataset. The authors now include a description of their general criteria in the text (lines 224-231) and repeated several times in their response, but the description lacks important details. Additional discussion of how soil depth was treated is also needed somewhere.

1) For specifics & justification (Line 224-231): "... the soil d15N data with the following conditions were excluded:

- (a) Soil depth > 30 cm;
- (b) Low C:N ratio: C:N ratio is too low to be considered as natural;
- (c) Low N concentration: N concentration is too low to be considered as natural;
- (d) Organic: C concentration is too high to be considered as mineral;
- (e) Marine: the sample site is adjacent to a marine ecosystem;
- (f) Litter: the sample is collected from the litter layer, and should not be considered as mineral."

For b, c, d: Exactly what C:N, %N, and %C values were "C:N ratio too low to be natural," "N concentration too low to be natural," and "C concentration too high to be mineral"? How low or high? Considered "natural" by what justification? For example, mineral subsoils (~20-30 cm) with little organic matter can have both low C:N ratios and low N concentrations. And, how is C from soil carbonates considered or excluded?

For e, please state why marine-adjacent ecosystems were excluded.

For f, please clarify if "litter layer" aims to encompass the entire organic horizon (when present),

or only its top layer – i.e., the L layer only or the whole O horizon (L+F+H)?

Combining d and f: it isn't apparent why the litter (or organic) layer is excluded from the estimates of soil d15N and resulting isotopic model f-denit calculations. As this layer is usually isotopically light compared to mineral soil, its exclusion is likely to increase soil d15N estimates, especially in high latitudes that have thicker organic horizons.

2) Re. data aggregation: how were d15N measurements aggregated over the specified 30 cm depth? Soil d15N usually increases with depth, and subsoil measurements are typically rare compared to surficial measurements. Were values computed as a simple average by grid cell, regardless of sampling depth, as long as < 30 cm? Or averaged by depth and then compiled with some sort of weighting? Presumably the latter approach was necessary, but either way, please describe how samples from different depths were treated when creating the gridded values.

Both sets of clarifications could go in the Supplement but are important to state somewhere, both to allow reproducibility of the analysis and for full interpretation of the resulting f-denit estimates. With these explanations added, I support publication in Nature Communications.

Responses to Reviewers

To Reviewer #1:

[Reviewer #1 General Comments]

The authors thought about my comments and addressed them well, including reevaluating data and conducting new analyses. The assumptions and approximations are more explicit and the paper is better. The one thing that I would like the authors to finally consider and provide is an explanation (can be brief statement or analysis) of the inference and mechanisms underlying the strong explanatory power of N fixer and EM abundance based on globally gridded data. The relative influences of these variables are strong in the models. Although the RF model should account for this, the global distributions of these symbioses are known to exhibit strong collinearity with Temp and other climate/latitudinal drivers. So what are the mechanisms? The description of these findings in results and supplemental are relatively muted, e.g. “Moreover, we would like to highlight the crucial roles of microsymbionts (Nfixer, ECM) and N deposition, which are potential variables for improving model...”. Without asking too much detail, please provide some depth here as to why these tree-microbe associations should imprint on soil isotope based denitrification estimates.

[Response] We thank the reviewer for the positive feedbacks and valuable comments on our revised manuscript. Following the reviewer’s comment, we have added the explanation of why microsymbionts play crucial roles in the prediction of global soil $\delta^{15}\text{N}$ map in Lines 266–269 in the revised manuscript:

“The crucial roles of microbial symbionts result from that the N fixing bacteria assimilates atmospheric N_2 into soil with its $\delta^{15}\text{N}$ signal close to zero, and the plants associated with ECM and AM have different pathways of N uptake from soil, with the isotope fractionation higher for ECM than AM^{33,34}.”

Similarly, we also added this explanation in Lines 78–81 in the Supplementary Text S1.

To Reviewer #2:

[Reviewer #2 General Comment 1]

1. I was Reviewer 2 on the prior version of this manuscript. I have reviewed the revised manuscript and the authors' response to review. I concur with the comments by Reviewer 1, and I commend the authors on both the initial analyses and for the substantial work invested in reviewer questions. The new Supplementary Figures 18 & 19 are fascinating, and provide important support for the main analysis. Although it would be nice for the main text to further discuss these findings on the direction and magnitude of key factors associated with soil $\delta^{15}\text{N}$, I'm glad the figures are at least in the Supplement.

[Response] We thank the reviewer for the positive feedback and approving our revised manuscript.

[Reviewer #2 General Comment 2]

2. I also like the table added per Reviewer 1's request (Supplementary Table 4), showing biome-level estimates of N inputs and f_{denit} . Could an additional row be added reporting the area of each biome? That addition would allow conversion between units of total N flux (Tg N/yr) and flux per unit area.

[Response] Thanks. Yes, we have added the area of each biome in Supplementary Table 4.

[Reviewer #2 General Comment 3]

3. Overall, the authors have nicely addressed my comments, although responses to one request from both reviewers raises even more questions: both Rev. 1 and I sought added explanation on how the authors used the Craine et al. soil ^{15}N dataset. The authors now include a description of their general criteria in the text (lines 224–231) and repeated several times in their response, but the description lacks important details. Additional discussion of how soil depth was treated is also needed somewhere.

[Response] Thanks. Following the reviewer's comment, we have 1) added the detailed criteria for excluding the soil samples in Lines 224–234; and 2) added the details of how multiple soil $\delta^{15}\text{N}$ measurements at different depths are treated in Lines 234–235. The detailed point-by-point responses are listed following the Detailed Comments as follows.

[Reviewer #2 Specific Comment 1]

1. For specifics & justification (Line 224-231): "... the soil $\delta^{15}\text{N}$ data with the following conditions were excluded:

- (a) Soil depth > 30 cm;
- (b) Low C:N ratio: C:N ratio is too low to be considered as natural;
- (c) Low N concentration: N concentration is too low to be considered as natural;
- (d) Organic: C concentration is too high to be considered as mineral;
- (e) Marine: the sample site is adjacent to a marine ecosystem;
- (f) Litter: the sample is collected from the litter layer, and should not be considered as mineral."

1) For b, c, d: Exactly what C:N, %N, and %C values were "C:N ratio too low to be natural," "N concentration too low to be natural," and "C concentration too high to be mineral"? How low or high? Considered "natural" by what justification? For example, mineral subsoils (~20-30 cm) with little organic matter can have both low C:N ratios and low N concentrations. And, how is C from soil carbonates considered or excluded?

2) For e, please state why marine-adjacent ecosystems were excluded.

3) For f, please clarify if "litter layer" aims to encompass the entire organic horizon (when present), or only its top layer – i.e., the L layer only or the whole O horizon (L+F+H)?

4) Combining d and f: it isn't apparent why the litter (or organic) layer is excluded from the estimates of soil $\delta^{15}\text{N}$ and resulting isotopic model f-denit calculations. As this layer is usually isotopically light compared to mineral soil, its exclusion is likely to increase soil $\delta^{15}\text{N}$ estimates, especially in high latitudes that have thicker organic horizons.

[Responses] We thank the reviewer for reminding the unclarity of data we used here. Following the reviewer's comments, we justified the detailed conditions for excluding the soil samples as follows:

1) Following the data use suggestion in Craine *et al.* (2015), the soil samples with C:N ratios < 1 gC gN⁻¹ are considered as "too low to be natural", with N concentration < 0.02 mg g⁻¹ as "too low to be natural", and C concentration > 610 mg g⁻¹ as "too high to be mineral".

2) Marine-adjacent sites were excluded from analysis because the soil $\delta^{15}\text{N}$ variations in marine-adjacent sites would be driven by mechanisms differing from those in terrestrial ecosystems since these sites involve a lot of N transformation processes in aquatic/coastal ecosystems (e.g., benthic N fixation, upwelling, diffusion, burial, and phytoplankton uptake) (Maren *et al.*, 2013; Liu *et al.*, 2021; Schafstall *et al.*, 2010).

3) We would like to clarify that the litter layer is only the top layer of soil column rather than the whole organic (O) horizon.

4) Sorry for the misleading text in the previous version, we would like to take this opportunity to clarify the sites ONLY with litter or O horizon are excluded in our study, because we consider soil samples with depth to 30cm, while litter or O horizon is less than 10 cm even in high latitude ecosystems. In other words, we included soil samples from 0 cm to 30 cm, which include both litter (O horizon) and top mineral layers. We ONLY excluded the sites that ONLY have litter or O horizon values, but do not have the below top mineral layers. In details, there are 973 observations located at litter (n=141) or O (n=832) layer, 95% of which have soil $\delta^{15}\text{N}$ at a depth less than 10 cm, and without soil $\delta^{15}\text{N}$ at mineral layers reported. Thus these shallow soil samples ONLY with O horizon should be excluded from our analysis to avoid bias between O horizon and O horizon + mineral layers.

To clearly and transparently justify the conditions for excluding soil samples, we added details for the data treatment in Lines 224–234 as follows:

“As the original soil $\delta^{15}\text{N}$ datasets from Craine *et al.*²⁸ cover multiple soil depths and contain soil samples from various sites, we used only the $\delta^{15}\text{N}$ data from soils with depth ≤ 30 cm, while $\delta^{15}\text{N}$ data with the following conditions were excluded: (a) soil depth > 30 cm; (b) C:N ratio is too low (<1 gC gN⁻¹) to be considered as natural; (c) N concentration is too low (<0.02 mg g⁻¹) to be considered as natural; (d) the sample is only collected from organic horizon without mineral layers, or C concentration is too high (>610 mg g⁻¹) to be considered as mineral; (e) the sample is only collected from litter layer, the top layer of the soil column; (f) the sample site is adjacent to a marine ecosystem, which may involve a lot of N transformation processes in aquatic/coastal ecosystems (e.g., benthic N fixation, upwelling, burial, and phytoplankton uptake)^{49–51}; (g) the sample is from cropland; (h) the sample is from pastures, drystocks, dairy and industrial sites.”

[Reviewer #2 Specific Comment 2]

2. Re. data aggregation: how were $\delta^{15}\text{N}$ measurements aggregated over the specified 30 cm depth? Soil $\delta^{15}\text{N}$ usually increases with depth, and subsoil measurements are typically rare compared to surficial measurements. Were values computed as a simple average by grid cell, regardless of

sampling depth, as long as < 30 cm? Or averaged by depth and then compiled with some sort of weighting? Presumably the latter approach was necessary, but either way, please describe how samples from different depths were treated when creating the gridded values.

[Response] Thanks for this valuable comment. The multiple soil $\delta^{15}\text{N}$ measurements within the depth of 30 cm were averaged weighted by the soil N content at different depths. Following the reviewer's comments, we have added this in Lines 234–235:

“The soil $\delta^{15}\text{N}$ within the depth of 30 cm were averaged weighted by soil N content if multiple depths were measured.”

[Reviewer #2 Specific Comment 3]

Both sets of clarifications could go in the Supplement but are important to state somewhere, both to allow reproducibility of the analysis and for full interpretation of the resulting f_{denit} estimates. With these explanations added, I support publication in Nature Communications.

[Response] We are grateful for your positive feedbacks on our revised manuscript. We have added the explanations and clarifications associated with the data treatment method in the revised manuscript.

References

Craine, J. M. *et al.* Convergence of soil nitrogen isotopes across global climate gradients. *Sci. Rep.*, **5**, 8280 (2015)

Maren, V., *et al.* The marine nitrogen cycle: recent discoveries, uncertainties and the potential relevance of climate change. *Phil. Trans. R. Soc. B*, **368**, 20130121 (2013).

Liu, X., *et al.* Simulated global coastal ecosystem responses to a half-century increase in river nitrogen loads. *Geophys. Res. Lett.*, **48**, e2021GL094367 (2021).

Schafstall, J., *et al.* Tidal-induced mixing and diapycnal nutrient fluxes in the Mauritanian upwelling region, *J. Geophys. Res.*, **115**, C10014 (2010).

Hartemink, A. E., *et al.* Soil horizon variation: A review. *Adv. Agron.*, **160**, 125-185 (2020).